



# Neighbouring time ensemble Kalman filter (NTEnKF) data assimilation for dust storm forecasting

Mijie Pang[1], Jianbing Jin[1], Arjo Segers[2], Huiya Jiang[3], Wei Han[4], Ji Xia[1], Li Fang[1], Jiandong Li[1], Hai Xiang Lin[5,6], and Hong Liao[1]

[1]Joint International Research Laboratory of Climate and Environment Change, Jiangsu Key Laboratory of Atmospheric Environment Monitoring and Pollution Control, Jiangsu Collaborative Innovation Center of Atmospheric Environment and Equipment Technology, School of Environmental Science and Engineering, Nanjing University of Information Science and Technology, Nanjing, Jiangsu, China
[2]TNO, Department of Climate, Air and Sustainability, The Netherlands
[3]College of Environment and Resources, Nanjing Agricultural University
[4]Numerical Weather Prediction Center, Chinese Meteorological Administration, Beijing, China
[5]Delft Institute of Applied Mathematics, Delft University of Technology, Delft, The Netherlands
[6]Institute of Environment Sciences, Leiden University, The Netherlands

**Correspondence:** Jianbing Jin (jianbing.jin@nuist.edu.cn)

**Abstract.** Dust storms pose significant threats to human health and property. Accurate forecasting is crucial for taking precautionary measures. Dust models have suffered from uncertainties from emission and transport factors. Data assimilation can help refine biased models by incorporating available observations, leading to improved analyses and forecasts. The Ensemble Kalman Filter (EnKF) is a widely-used assimilation algorithm that effectively tunes models, particularly in terms of intensity

5   adjustment. However, when the position of the simulation does not align consistently with the observations which is referred to as position error, the EnKF algorithm struggles. This is because its background covariance normally represents intensity uncertainty, while the positional errors in the long distance transport are difficult to be quantified and were usually neglected. In this paper, we propose a novel Neighboring Time Ensemble Kalman Filter (NTEnKF). In addition to the original ensembles quantifying dust loading variation, this methodology introduces extra ensembles from neighboring time for describing the potential spread of dust position. The enlarged ensemble captures both intensity and positional errors, allowing observations to

10  be thoroughly resolved into the assimilation calculations. We tested this method on three major dust storm events that occurred in spring 2021. The results show that position error significantly degraded dust forecasting in terms of RMSE and NMB, and hindered the EnKF from assimilating valid observations. In contrast, the NTEnKF yielded substantial improvements in both dust analysis fields and forecasts compared to the EnKF.

## 15   1   Introduction

Dust storms are a natural meteorological disaster (Zhang et al., 2005), whose occurrence is attributed to frequent strong winds over dry and loose soil texture (An et al., 2018). Dust particles can be lifted up to a few miles and transported over thousands of kilometers away (Zhang et al., 2018), with dust aerosol concentrations as high as thousands of $\mu$g m$^{-3}$ (She et al., 2018).



Meanwhile, these aerosols can interact with SO$x$ and NO$x$ undergoing heterogeneous chemical reactions during transportation, leading to further severe aerosol pollution (Song et al., 2022). These pose a great threat to human health by causing damage to the respiratory and circulatory systems (Gross et al., 2018; Goudie, 2014). East Asia, as one of the major dust sources and affected regions (Hu et al., 2019), has drawn much attention from researchers. For instance, in the 2021 spring, several super dust storms, which are recorded as the largest ones in terms of intensity and coverage in a decade (Filonchyk and Peterson, 2022), swept over East Asia and caused huge loss of lives and properties both in Mongolia and China (Gui et al., 2022; Jin et al., 2022; Tang et al., 2022). An accurate early warning of dust storms is, therefore, in essential need to help minimize the damages.

The growing interest in dust storms from the public has stimulated the understanding of the physical processes associated with the dust cycles over the past decades. To achieve the simulation of dust storms, several dust emission parameterization schemes have been proposed since the early 1990s, e.g., MB95 (Marticorena and Bergametti, 1995), Shao96 (Shao et al., 1996; Shao, 2004), Zender03 (Zender et al., 2003), and K14 (Kok et al., 2014). Coupled with chemical transport models, dust simulations could then be carried out, e.g., CUACE/DUST (Chinese Unified Atmospheric Chemistry Environment for Dust) (Gong and Zhang, 2008), BSC-DREAM8b (Dust Regional Atmospheric Modeling) (Pérez et al., 2006; Mona et al., 2014), GEOS-Chem (Duncan Fairlie et al., 2007), and LOTOS-EUROS (Timmermans et al., 2017; Manders et al., 2017). These dust models help evaluate health effects, quantify Earth system impacts, and reveal the synoptic climatic driving forces, and also to build dust early warning systems via reporting the dust loading in the few hours to few days. However, various numerical approximations are used to solve the dynamic dust equations, so that the model configuration (like coarse grid cell and time step), uncertain input data (e.g., wind field and boundary/initial conditions) inevitably limit the model forecast skill (Mallet and Sportisse, 2006). Notably, it is widely accepted that uncertainty in the emission parameterization is the largest error source of dust simulation (Ginoux et al., 2001, 2012; Di Tomaso et al., 2017, 2022; Jin et al., 2019a, b). The performance of numerical dust models degrades greatly due to these factors.

Observation is another fundamental method for exploring the intensity and spatial distribution of dust storms (Muhammad Akhlaq et al., 2012). Satellite-based observations are a rapidly developing technology that is widely used in detecting dust storms (Gui et al., 2022). Products from satellites such as MODIS, Himawari, and Fengyun-4A provide various information about aerosol properties with high spatial resolution and extensive coverage. However, they only retrieve column-cumulative values and are easily affected by clouds and other particles. Therefore, significant uncertainties and biases exist, and pre-processing is necessary before they can accurately represent dust load (Jin et al., 2019b, 2022). Ground-based observation networks, on the other hand, are highly reliable and have high temporal resolution, making them indispensable for measuring dust aerosol concentration (She et al., 2018). In recent years, China has invested heavily in the construction of a ground station network, and there are now over 1600 ground stations throughout China that provide a comprehensive picture of dust plumes (Gui et al., 2022). The national observation network provides rich measurements for investigating dust storms in East Asia.

Data assimilation is a powerful technique that integrates models and observations. Based on Bayesian theory, data assimilation algorithm is intended to calculate the posteriori probability distribution of the model state given the observations as accurately as possible (Law and Stuart, 2012). Two main approaches to data assimilation are variational methods and filtering





methods. Variational methods, such as 4DVar, aim to retrieve an optimal posterior analysis that fits both the prior and mea-
surements over a time window by minimizing a cost function (Rabier and Liu, 2003). Variational methods are widely used in
inverse modeling of initial conditions and emission fields (Jin et al., 2022; Bergamaschi et al., 2010; Corazza et al., 2011) and
reanalysis data, but they require tangent linearization or adjoint of the model, which can be challenging to develop and main-
tain. The cost function minimization is computationally demanding, especially for high-dimensional and nonlinear models.
Filtering methods, on the other hand, assimilate observations sequentially and are more efficient for operational forecasting
systems. Various filtering approaches, such as Kalman Filter, Extended Kalman Filter, and Particle Filter, have been developed.
Among all the filtering methods, the Ensemble Kalman Filter (EnKF) is the most popular filtering method due to its ability to
handle high-dimensional models, easy parallelization (Evensen, 1994). EnKF has been successfully applied in various disci-
plines, e.g.,weather forecasting (Houtekamer et al., 2005) and hydrology (Reichle et al., 2002). Its advantages include handling
nonlinearity and non-Gaussian observations, not requiring explicit calculation of tangent linear operators, and computational
efficiency. EnKF uses a limited ensemble to approximate the background covariance statistics of the model, making it highly
efficient for operational forecast models.Overall, EnKF can be applied in various fields especially in atmospheric models to
improve the accuracy of forecast and to better understand complex systems.

The primary source of uncertainty in dust simulation is related to the online emission parameterization. Therefore, most
previous studies on dust storm data assimilation have focused on emission inversion. For example, Yumimoto and Takemura
(2015) used MODIS AOD retrievals for long-term dust emission inverse modeling over Asia. Escribano et al. (2017) inves-
tigated the impact of five different satellite AOD products on dust emission inversion over northern Africa and the Arabian
Peninsula. Their results indicated that the assimilation outcome is more sensitive to model uncertainties than to observational
uncertainties in some cases. The uncertainties in model actually have a greater impact in the assimilation results. In recent
studies, we have carefully explored the variability of dust emission over the Mongolia and China Gobi desert by assimilating
ground-based $PM_{10}$ concentration (Jin et al., 2018), polar-orbiting MODIS (Jin et al., 2022), and geostationary Himawari-8
AOD measurements (Jin et al., 2019b). To effectively improve dust storm emission inversion, we introduced observation bias
correction (Jin et al., 2019a), adjoint-based emission source tracking (Jin et al., 2020), and grid distortion (Jin et al., 2021).
These works provide valuable insights into the dynamics of dust emission and quantify their impacts on the environment
and climate. However, little attention has been paid to the application of high-quality dust storm sequential forecasting us-
ing filter methods. Recently, we have developed a data assimilation-based operational dust forecasting system by coupling
Ensemble Kalman Filter (EnKF) and Localized Ensemble Kalman Filter (LEnKF) assimilation algorithms with a chemical
transport model (LOTOS-EUROS) through an interface of our self-designed assimilation toolbox, Pyfilter (Pang, 2023). We
tested this system on the super dust storms that occurred in the spring of 2021, as we will show later. Significant improvements
were found in the assimilation analysis and assimilation-based forecasts compared to the pure model results. Furthermore, the
LEnKF algorithm with a proper localization distance threshold was consistently shown to be superior to the EnKF algorithm.

Despite the positive results obtained from our tests, there are still unresolved errors. One major issue is the apparent mismatch
between the observations and model in space after long-distance transport. In addition to the discrepancy in the dust intensity,
as will be illustrated in Sect. 2.4, the timing of the dust arrival and departure reported by the model simulation also differs



heterogeneously from reality. The dust intensity is a key feature, as well as the position, when evaluating a dust forecast.

The former represents the actual dust load, while the latter reveals where the dust plume affects at a given instant. For an operational forecasting and warning system, the position information is sometimes more important than the intensity. In terms of mathematical metrics, such as root mean square error, the forecasting skills degraded significantly with the presence of the position mismatch. The detailed mechanism behind this issue and its further consequences will be illustrated in Section 3.2.

The so-called "position error" in dust aerosol simulations typically arises after long-distance transport. There are many

factors that contribute to the position error, such as simplified physical processes, coarse model resolution, uncertain physical parameters (Ravela et al., 2007), and the uncertainty in the meteorological field and emission timing, as illustrated in our previous work (Jin et al., 2021). Similar to the dust emission inversion studies discussed above, the deviations between the model and observations in dust storm data assimilation are also attributed to the uncertainty in the dust emission, where ensemble individuals are generated with perturbed dust emission fields. However, the uncertainty in the dust plume position

is difficult to quantify and is hardly taken into account when designing the background error covariance of the simulated dust plume. Therefore, classic dust data assimilation methodologies now focus on intensity adjustment and are not capable of handling the imbalanced uncertainties between the observations and simulation caused by the position error.

Position error is not a mere occasional issue, but rather a cumulative error that accompanies model simulations over time. This type of error is quite common in forecasting phenomena such as hurricanes, dust storms, thunderstorms, and precipitation

(Dance, 2004; Nehrkorn et al., 2015; Jin et al., 2021). However, there have been relatively few studies aiming to address this problem. Brewster (2003) proposed an objective method for identifying and correcting position errors using densely-distributed and high-resolution observational data. Their research demonstrated that it is possible to correct position errors in Observing System Simulation Experiments (OSSEs). Jin et al. (2021) developed a grid-distortion technique based on image morphing and post-processing, which successfully realigned dust plumes to better match the measurements. Both of these improvements rely

on densely distributed observations, but often the observations do not fully cover the entire domain, limiting the applicability of these methods.

In this paper, we introduce a novel Neighboring Time Ensemble Kalman Filter (NTEnKF) by incorporating an ensemble member resampling strategy into the EnKF algorithm. For assimilation analysis at a given time, the background error covariance of the simulated dust plume is calculated using not only the original ensemble simulation with perturbed emissions, but

also the same ensemble simulations at neighboring moments (a few hours earlier and later). These additional ensemble members represent the potential position spread of the actual dust plume, effectively accounting for transport errors. The resampled ensemble members quantify the complex covariance that captures both intensity and position error dynamics, without requiring additional processing on observations, meteorological fields, or other physical parameters. We tested the NTEnKF on three severe dust storm events that occurred in 2021. Our results show superior assimilation performance compared to the standard

EnKF, particularly when position errors are present in the simulated dust plume.

This paper is organized as follows: Section 2 introduces the dust measurements and dust model used in the research. We also discuss that the major uncertainty of dust model forecast comes from the emission. But there is another problem: position error that remains to be solved. Then in Sect. 3, we explain introducing the procedure of ensemble-based assimilation algorithm





and the mechanism of position error's negative effect on EnKF. How the new assimilation method works is explained in detail

afterwards. To test the performance of NTEnKF, sequential assimilation experiments on several dust storm events are designed. Section 4 analyses the results of experiments in terms of both the assimilation analysis and forecast performance. Section 5 concludes this paper.

## 2 Dust measurement, model and position error

In this paper, ground-based $PM_{10}$ is used as the measurement with a bias-correction procedure to remove the non-dust part. The

dust model adopted is the LOTOS-EUROS. Considering the model processes, the greatest uncertainty in the dust simulation comes from uncertainty the emission parameterization. Meanwhile, uncertainties from meteorology can also influence the model forecast and lead to the "Position error".

### 2.1 Dust measurements

Thanks to the continuous efforts and investments from the Ministry of Ecology and Environment, over 1600 ground monitoring

stations have been established across China, with some locations in northern China shown in Fig. 1. These stations provide real-time hourly air quality data, and their hourly $PM_{10}$ concentrations serve as indispensable datasets for measuring dust load, which are used as observations in this paper.

Despite the advantages of low uncertainty and high time resolution, $PM_{10}$ observations are not assimilated directly due to the mixed state of dust and non-dust aerosols in the original $PM_{10}$ data. Anthropogenic activities, such as vehicle emissions,

coal burning, and industrial processes (Wu et al., 2018; Liu et al., 2018), along with natural sources like volcanic eruptions, sea spray, wildfires, and wind-blown dust contribute to the total $PM_{10}$ concentration. Assimilating $PM_{10}$ data directly into a dust model may introduce biased errors and lead to model divergence (Jin et al., 2019a). Therefore, it is necessary to eliminate the bias before data assimilation.

In this study, the non-dust portion of $PM_{10}$ is approximated through a separate model. The dust observations assimilated are

calculated by subtracting the non-dust fraction from the original $PM_{10}$ measurements. Further details regarding the baseline removal (BR) can be found in Jin et al. (2022).

### 2.2 Dust model

In this paper, the LOTOS-EUROS v2.1 is used to simulate dust storms that occurred in East Asia. Originating from the Long-Term Ozone Simulation (LOTOS) and the European Operational Smog model (EUROS) in the 1980s, LOTOS-EUROS has

undergone continuous development for various applications. It has been widely used in air quality forecasting (Curier et al., 2012; Brasseur et al., 2019; Lopez-Restrepo et al., 2020; Skoulidou et al., 2021), dust/aerosol emission inversion (Yarce Botero et al., 2021; Jin et al., 2018, 2019a, b, 2021, 2022), and source apportionment (Kranenburg et al., 2013; Timmermans et al., 2017; Pommier et al., 2020; Jin et al., 2020). In spring 2021, three major dust storm events occurred in East Asia, around 15th March, 28th March, and 15th April. These events, referred to as DSE1, DSE2, and DSE3, are used as test cases in this study.





These dust storms caused significant losses in both Mongolia and China (Jin, 2021; Chen and Walsh, 2021). Accurate forecast of such severe sandstorms is crucial for reducing health and property damages.

To simulate the dust storm over East Asia, LOTOS-EUROS is configured following our recent work (Jin et al., 2022): The simulation domain is from $15°$ N to $50°$ N and $70°$ E to $140°$ E with a grid resolution of $0.25° \times 0.25°$. The model consists of 8 layers with a top at 10 km. The boundary conditions are set to zero assuming that all the dust aerosols are emitted during the

simulation window. Dust emission, deposition, advection, diffusion and dry/wet deposition are considered within the model. The model output is at the interval of 1 hour. The meteorological field used in the model is from European Center for Medium-Ranged Weather Forecast (ECMWF) operational forecast over 3-12 hours. Its grid resolution is about 7 km. The 3-hourly short-term meteorological forecast field is interpolated to hourly values. The grid resolutions are also averaged to fit the model resolution.

The whole model simulation period is set from 13 to 17 March for DSE1, 27 to 30 March for DSE2 and 14 to 17 April for DSE3, which covering the whole life cycles of emission and long-distance transport. More details could be found in Jin et al. (2022).

## 2.3 Emission uncertainty

The goal of this study is to calculate the dust concentration field that best fits both the a priori and observations at each

assimilation analysis. The optimized field will then be used as the initial condition for sequential dust forecasts, as explained in Section 3.1. It is essential to define and quantify the uncertainty in dust simulations. As previously mentioned, the uncertainty in emission parameterization is widely believed to be the dominant error source in dust simulation (Ginoux et al., 2001, 2012; Di Tomaso et al., 2017, 2022; Jin et al., 2019a, b). High levels of uncertainty in dust emission parameterization arise from insufficient knowledge about windblown erosion, lack of accurate input on soil characteristics, and the models' inability to

resolve the fine-scale variability in wind fields governing dust emission (Escribano et al., 2017; Foroutan et al., 2017; Foroutan and Pleim, 2017; Jin et al., 2019b).

In our recent work (Jin et al., 2022), a 4DVar-based inverse modeling approach was employed to retrieve an optimal emission field for the three major dust storms in spring 2021 (Jin et al., 2022). The a priori emission, $f_{\text{priori}}$, followed the *Zender03* dust emission parameterization scheme (Zender et al., 2003). To compensate for potential errors, a spatially varying multiplication

factor was introduced. Mathematically, it was quantified by a background error covariance matrix, $\mathbf{B}$, to describe the potential spread of the actual dust emission flux.

In this study, we assign the dust simulation uncertainty to the emission error as well. Ensemble emission field $[f_1, ..., f_N]$ are generated randomly following the emission uncertainty choice $f_{\text{priori}}$ and $\mathbf{B}$ in Jin et al. (2022). They are used to forward the LOTOS-EUROS model $\mathcal{M}$ for the ensemble dust simulations $[x_1, ..., x_N]$ as:

$$[x_1, ... , x_N] = [\mathcal{M}(f_1), ... , \mathcal{M}(f_N)] \tag{1}$$

N refers to the total ensemble number. In this paper, 32 ensembles is used.





These ensemble individuals are used in the EnKF assimilation for representing the covariance dynamics of the dust plume, which resulted in more accurate dust analysis and forecast as will be shown in Sect. 4. However, the ensemble realizations mainly represent the uncertainty in the intensity feature, and hardly help resolve the positional deviation between the observa-

tion and simulation. The presence of position error would give rise to a divergent assimilation analysis as will be illustrated in Sect. 3.1.

### 2.4  Position error

For all the three dust events, most of the dust particles were originated from the Mongolia Gobi desert, and carried by the prevailing wind towards southeast. After several thousands of kilometers transport which lasted about one to two days, they

finally arrived in the densely-populated northern China.

Position errors are clearly visible in the simulation of the first two dust events (DSE1 and DSE2). Examples can be best seen in Fig. 1, which plots the evolution of LOTOS-EUROS simulated surface dust concentration alongside BR-PM$_{10}$ (BR: non-dust baseline-removed) concentration observations for DSE1 (panel a) and DSE2 (panel c). The corresponding standard deviations from ensemble model simulations and the model-minus-observation differences (absolute values) are also plotted in panel b

and panel d. In panel a.1, the model generally simulates a similar shape of the dust plume as indicated by the observations at the first instance, though the dust load intensities differ to some extent. However, during the subsequent transport, positional errors arise gradually. For example, in panel a.2, the right part of the simulated dust plume is positioned about 100 to 200 km too far south compared to ground-based observations. Consequently, the Root Mean Square Error (RMSE) increases significantly from 594 $\mu$g m$^{-3}$ at 8:00 to 831.5 $\mu$g m$^{-3}$ at 11:00. This position error continues to accumulate over the following 3 hours at 14:00.

The development of position errors is further clearly visible against the PM$_{10}$ observations, especially in the enlarged green box in panel a.3. The model simulation missed all the dust load there, while the observations indicate a significant amount of dust aerosols. It can also be seen in panel b.3 that the model-minus-observation differences exceed 1000 $\mu$g m$^{-3}$ there. Similarly, for DSE2 occurring on 28th March, 2021, as shown in Fig. 1(c), discrepancies between observations and simulation become more explicit as time evolves, especially for the dust in the light blue box in panels c.1 and c.2. The RMSE increases

from 490.1 $\mu$g m$^{-3}$ at 5:00 to 650.5 $\mu$g m$^{-3}$ at 8:00, and this error expands to a wider extent as shown in the enlarged green box in panel c.3. This position error not only limits the pure model forecast performance but also significantly degrades the subsequent assimilation analysis and forecast. With an ensemble-approximated background covariance focusing on intensity error, neither the position deviation nor the intensity deviation can be fully resolved, as will be explained in Sect. 3.2.

Potential sources of position error in dust model may be attributed to inaccuracies in emission timing, uncertainties in mete-

orological input data (e.g., wind fields responsible for transporting dust plumes from the Gobi Desert in Mongolia and China to downwind regions), or a combination of these factors. Adjusting the emission timing profile, which characterizes the release of soil particles into the atmosphere, could partially correct the position of the dust plume. Moreover, alterations in meteorological conditions governing long-distance transport might also realign the dust plume's position. To address the position error, a comprehensive covariance matrix is necessary to account for both the potential variations in emission temporal profiles

and the accumulation of uncertainties along the plume's extensive trajectory. Concurrently, a significantly larger ensemble size



is required to propagate these uncertainties, featuring high degrees of freedom, into the $PM_{10}$ observational space. Although a sophisticated covariance matrix and a substantial ensemble size (resulting in considerable computational cost) may aid the EnKF in simultaneously resolving position and intensity errors, this approach is often prohibitively expensive. Therefore, an efficient and cost-effective alternative solution is required.

**Figure 1.** Evolution of the simulated dust plume from average of 32 model ensembles with scatter of ground BR-$PM_{10}$ observations (**a.1-3**). Their corresponding standard deviation from model ensembles with scatter of the model-minus-observation differences (absolute value) (**b.1-3**) at 08:00, 11:00 and 14:00 15th March, 2021, respectively. Figures below are the same except the time is at 05:00 (**c.1** and **d.1**), 08:00 (**c.2** and **d.2**), 11:00 (**c.3** and **d.3**) 28th March, 2021, respectively. BR-$PM_{10}$: baseline-removed $PM_{10}$. The colorbar in panel **a** and **c** represents the concentrations, and the colorbar in panel **b** and **d** represents the model-minus-observation differences (left) and standard deviation (right).



## 3 Assimilation methodology and experiments

EnKF is a powerful algorithm to tune the model simulation with observations especially in intensity adjustment given the perturbed emission spreads. However, when faced with the position error, its weakness is exposed that some model-minus-observation inconsistency cannot be resolved by EnKF as illustrated in Section 3.1. On the contrary, our NTEnKF can correct both the position error and the intensity. Assimilation strategy is designed and embedded into a assimilation forecast system in Section 3.2. Experiments are designed on the dust storms occur in spring, 2021, which are illutrated in Section 3.3.

### 3.1 EnKF

The Ensemble Kalman Filter (EnKF) was first proposed by Evensen (1994). Stemming from the Kalman Filter, it was designed to address high-dimensional problems by employing limited ensembles to approximate the true background error covariance. The EnKF has been proven to be practical and efficient in various applications, particularly in sequential forecasting with the aid of localization (Lopez-Restrepo et al., 2020; Park et al., 2022). In any sequential forecast system, the objective of assimilation analysis is to provide an optimized initial state or parameter field, which, in this study, corresponds to the 3D dust concentration. This is achieved by assimilating the available measurements. The estimated dust concentration field can then be used to onward the model for more accurate dust forecasting.

Here is how the EnKF analysis works: starting from the priori dust concentration field $\boldsymbol{x}_t^{f,i}$ at time $t$ which is calculated by model integral operator $\mathcal{M}$ from the dust concentration field at the previous time step $\boldsymbol{x}_{t-1}^{a,i}$.

$$\boldsymbol{x}_t^{f,i} = \mathcal{M}(\boldsymbol{x}_{t-1}^{a,i}) \tag{2}$$

$$\mathbf{X}^f = [\boldsymbol{x}_t^{f,1}, \boldsymbol{x}_t^{f,2}, \cdots, \boldsymbol{x}_t^{f,\mathrm{N}}] \tag{3}$$

Note that for the first analysis the prior dust simulation are extracted from the model with the perturbed emissions as shown in Eq. 1. The $i$ represents the ensemble individual number. N is the number of ensembles. $\mathbf{X}^f$ is the ensemble model simulation matrix consists of the whole ensemble individuals.

The ensemble perturbation matrix $\mathbf{X}^{f\prime}$ calculates the deviation between the ensemble individuals $\boldsymbol{x}_t^{f,i}$ and the ensemble mean state $\overline{\boldsymbol{x}}_t^f$.

$$\overline{\boldsymbol{x}}_t^f = \frac{1}{\mathrm{N}} \sum_{i=1}^{\mathrm{N}} \boldsymbol{x}_t^{f,i} \tag{4}$$

$$\mathbf{X}^{f\prime} = [\boldsymbol{x}_t^{f,1} - \overline{\boldsymbol{x}}_t^f, \boldsymbol{x}_t^{f,2} - \overline{\boldsymbol{x}}_t^f, \cdots, \boldsymbol{x}_t^{f,\mathrm{N}} - \overline{\boldsymbol{x}}_t^f] \tag{5}$$

Then the background error covariance matrix $\mathbf{P}^f$ is approximated by $\mathbf{X}^{f\prime}$ as follows:

$$\mathbf{P}^f = \frac{1}{\mathrm{N}-1} \mathbf{X}^{f\prime} \mathbf{X}^{f\prime\,\mathrm{T}} \tag{6}$$





Afterwards, the Kalman gain $\mathbf{K}$ can be calculated with $\mathbf{P}^f$ and $\mathbf{O}$.

$$\mathbf{K} = \mathbf{P}^f \boldsymbol{\mathcal{H}}^{\mathrm{T}} (\boldsymbol{\mathcal{H}} \mathbf{P}^f \boldsymbol{\mathcal{H}}^{\mathrm{T}} + \mathbf{O})^{-1} \tag{7}$$

$\mathbf{K}$ weights the increments given from the observations to the priori estimation. In this paper, they are the BR-PM$_{10}$ observations stored in $\boldsymbol{y}$ and dust simulation stored in vector $\boldsymbol{x}$. $\boldsymbol{\mathcal{H}}$ is the observation operator which maps the model states into the observational space.

$\mathbf{O}$ is the observational error covariance matrix that weights the uncertainty of the measurements. In this case, it is the uncertainties from ground-based BR-PM$_{10}$ concentrations. $\mathbf{O}$ is defined as follows: the minimum uncertainty threshold is set to be 200 $\mu$g m$^{-3}$. Root of error from observations below the threshold is set to be 200 $\mu$g m$^{-3}$ and those over it is set to be 200+($\boldsymbol{y}$-200)$\times$0.2 $\mu$g m$^{-3}$. This definition can prevent the posteriori from getting too close to the low value observations and thus leading to model divergence. $\mathbf{O}$ is a diagonal matrix assuming that all the observations are independent.

In the end, the posteriori estimation individual $\boldsymbol{x}_t^{a,i}$ can be updated as follows:

$$\boldsymbol{x}_t^{a,i} = \boldsymbol{x}_t^{f,i} + \mathbf{K}(\boldsymbol{y} + \boldsymbol{\epsilon}^i - \boldsymbol{\mathcal{H}} \boldsymbol{x}_t^{f,i}) \tag{8}$$

$\boldsymbol{\epsilon}^i$ represents the sampling error vector. It is a random vector subjecting to normal distribution. Its mean is 0 and covariance is the root of diagonal from $\mathbf{O}$.

The equations presented above describe the Ensemble Kalman Filter (EnKF) algorithm for dust storm assimilation, which focuses on intensity adjustment. The EnKF assimilation aims to compute an optimal posteriori estimation given a priori information and observations. It is highly dependent on the observations and the ensemble spread. In fact, the ensemble-based background covariance matrix, $\mathbf{P}^f$, utilizes the ensemble members to approximate the true background covariance. The spatial distribution of the standard deviation (square root of the diagonal values in $\mathbf{P}^f$) from 32 model ensembles, along with the scatter of absolute model-minus-observation differences in two cases (DSE1, DSE2), is shown in Fig. 1 (b,d). In general, their spatial distribution corresponds well to the simulated dust field depicted in Fig. 1 (a, c). Concurrently, the uncertainty in the light blue box decreases rapidly as the simulated dust plume moves southward, as illustrated in panels b.1 and b.2. This suggests that our ensemble model simulations are highly confident that there are less affected by dust aerosols. However, the observations indicate that this area remains heavily polluted. In the case of DSE2, the situation becomes more complex. The simulated dust plume in DSE2 covers most of the observation area with a high dust load, as demonstrated in panels c.1 and d.1. The uncertainty, on the other hand, reveals that the ensemble model is less confident about the dust load, especially in the light blue box displayed in panel d.2. After 3 hours, these discrepancies become more evident. The extent to which this situation affects the EnKF assimilation will be discussed in this paper. It poses a challenge to EnKF assimilation in resolving the high-value measurements in this region.

The performance of EnKF deteriorates when position errors are present. The underlying mechanism can be best understood by examining Fig. 2(a). At time point $t_0$, there are ensemble model simulations (gray dashed lines) distributed across the three-dimensional space. The black line and blue star represent the average of model ensembles and observations, respectively. As clearly depicted, there is a positional mismatch between the ensemble model simulations and observations. Following the





assimilation analysis, the intensity of the dust plume is adjusted to better match the observations. However, in the spatial domain outside the priori, the dust concentration is reduced to near-zero levels. The observations in this area, containing valuable information about dust load, contribute little to correcting the dust load. This is due to the unanimous agreement
on the dust load from the model ensembles, which represents low uncertainty. In such cases, the assimilation analysis favors the model results and disregards the observations. Consequently, the a posteriori estimate is biased as a result of imbalanced uncertainties.

## 3.2 Neighboring time EnKF (NTEnKF)

To efficiently perform the assimilation analysis with both the intensity and position errors present, we propose a novel EnKF-
295 based method: Neighboring time Ensemble Kalman Filter (NTEnKF). The strategy is illustrated in Fig. 2(b). Instead of using the ensemble simulations solely at the exact assimilation analysis instant $t_0$, as shown in panel a, ensembles at neighboring moments are also introduced to expand the ensemble group. These resampled ensembles at neighboring times represent the potential positions of the actual dust plume. The enlarged ensembles exhibit a more extensive spread of the dust plume in the spatial domain compared to those displayed in panel a. The joint ensemble model simulations then capture uncertainty in both
intensity and position. The a posteriori estimate (red line) is adjusted to better fit the observations, with both of these errors resolved.

Mathematically, the NTEnKF procedures are very similar to those of EnKF, except that the original $\mathbf{X}^f$ is replaced by $\mathbf{X}^{f,new}$, which stores the enlarged ensemble members at the assimilation analysis instant and neighboring times. It starts with

$$\mathbf{X}^{f,new} = [\boldsymbol{x}_{t-1}^{f,1}, \boldsymbol{x}_{t-1}^{f,2}, \cdots, \boldsymbol{x}_{t-1}^{f,N}] + ...[\boldsymbol{x}_t^{f,1}, \boldsymbol{x}_t^{f,2}, \cdots, \boldsymbol{x}_t^{f,N}] + [\boldsymbol{x}_{t+1}^{f,1}, \boldsymbol{x}_{t+1}^{f,2}, \cdots, \boldsymbol{x}_{t+1}^{f,N}] \tag{9}$$

Let $t_0$ be the exact assimilation time, then $t_{-1}$ represents the time in the past, and $t_{+1}$ represents the time in the future. It is noteworthy that the time axis, denoted by $t_{-1}$ and $t_{+1}$, is utilized solely to illustrate the application of ensemble simulations at different time direction in the formula. However, in practical applications, ensembles from multiple adjacent time instants can be incorporated, as demonstrated in the horizon choice utilized in this study (as presented in Table 1).

Subsequently, the ensemble-based background covariance $\mathbf{P}$, Kalman gain $\mathbf{K}$ and posteriori state $\boldsymbol{x}^a$ will be updated with
310 the $\mathbf{X}^{f,new}$ in Eq. 6 $\sim$ 8, respectively.

The localization method is also adopted here to cut off the spurious correlation in $\mathbf{P}^f$ and constrain the background covariance to a certain distance. The localization matrix is constructed following Gaspari and Cohn (1999) (Eq. A.27) with a distance threshold $L_{thres}$. The details about the construction of $\mathbf{L}$ can be found in Supporting Information. The localized $\mathbf{P}^{f,local}$ is obtained by point to point multiply with localization matrix $\mathbf{L}$.

$$\mathbf{P}^{f,local} = \mathbf{P}^f \circ \mathbf{L} \tag{10}$$

With the localized $\mathbf{P}^{f,local}$, the localized posteriori estimation $\boldsymbol{x}_t^{a,i}$ can be updated via Eq. 7 and Eq. 8.

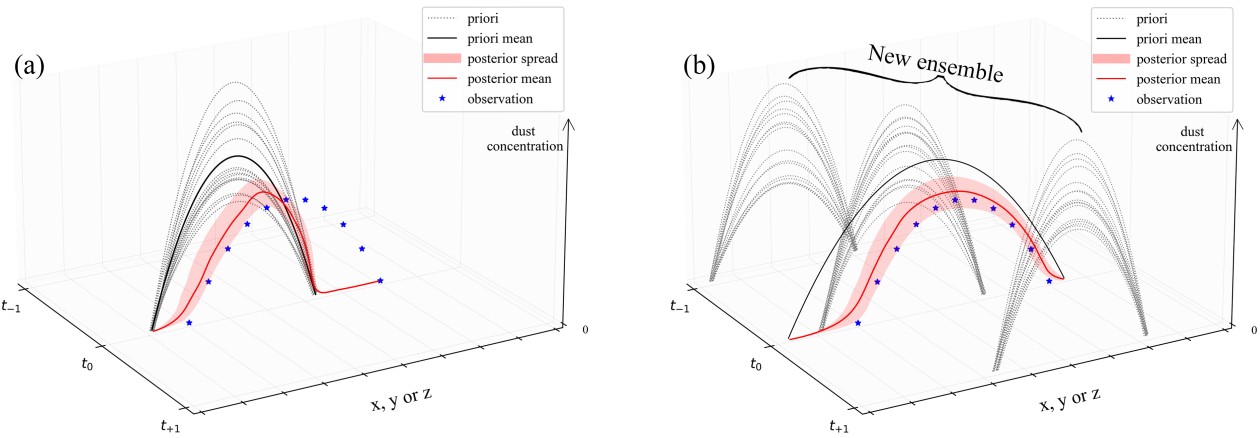

**Figure 2.** Strategy illustration of Ensemble Kalman Filter (EnKF) (**a**) and Ensemble Kalman Filter with neighboring time resampling (NTEnKF) (**b**). Figure axis left represents the time and right represents the position of the dust field in 3D space. The vertical axis represents the intensity of the dust.

### 3.3 Assimilation experiments

Both the EnKF and NTEnKF described above are embedded into our self-designed assimilation toolbox, PyFilter (Pang, 2023). This toolbox features a flexible interface for linking to numerical models (Pang et al., 2023), such as the dust storm forecasting model LOTOS-EUROS used in this study.

To evaluate the performance of the NTEnKF-implemented dust storm forecasting system, data assimilation experiments are conducted on three spring dust events in 2021. Five experiment sets are designed, as shown in Table 1. *Control run* represents the pure model forecast throughout the entire dust storm period. *EnKF* and *L500* denote the assimilation-based forecasts by EnKF and localized EnKF (LEnKF) with a localization distance threshold of 500 km, respectively. *NTEnKF* and *NTL500* represent the assimilation-based forecasts by NTEnKF and NTEnKF with a localization distance threshold of 500 km. Note that various distance thresholds have been tested for localization, and a choice of 500 km is found to provide the optimal assimilation analysis and forecast in our tested cases. The metrics, Root Mean Square Error (RMSE) and Normalized Mean Bias (NMB), are employed in this paper to evaluate system performance.

The first assimilation analysis did not commence until the dust plume was detected by the ground-based observation network and a position mismatch emerged. Three sequential EnKF analyses are conducted in each dust event at three-hour intervals. The timeline for DSE1 and DSE2 is depicted in Fig. 3.

Taking DSE1 as an example, the initial assimilation analysis is performed at 11:00 March 15, when an apparent position error was present, as illustrated in Fig. 1 (a.2). The last analysis is carried out at 17:00 March 15. As the dust loading decreases





rapidly when the plume moves further southeast, no additional assimilation is performed. A rolling forecast (red line with arrow) is generated based on the optimized dust concentration field with a 24-hour horizon for the purpose of examining forecast skill. Although no apparent position error is captured in the model forecast for DSE3, it is also tested with EnKF and NTEnKF, and the results are provided in the Supporting Information. The initial assimilation time for DSE3 is set to 20:00, and rolling forecasts are generated in the same manner as for DSE1 and DSE2.

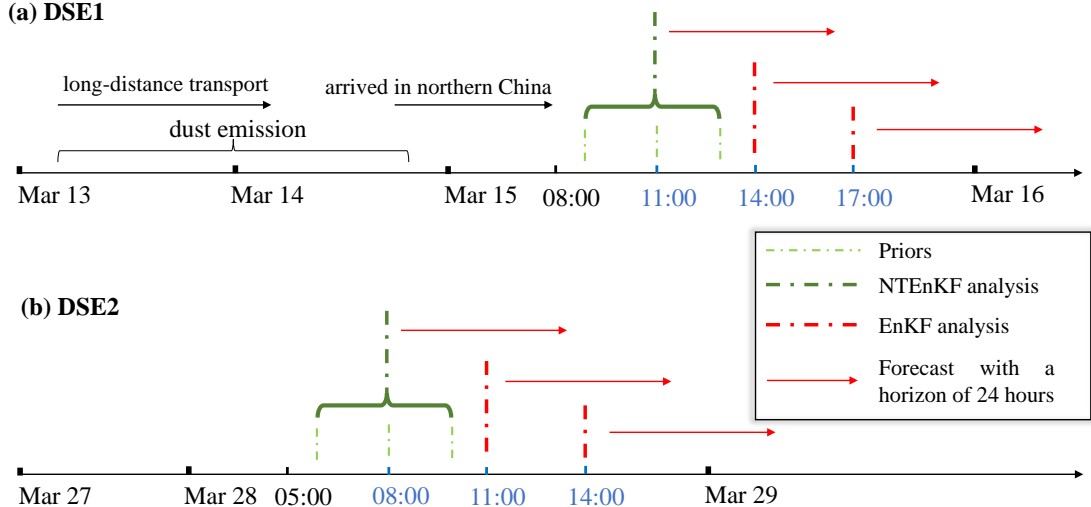

**Figure 3.** Sequential assimilation time set for DSE1 (**a**) and DSE2 (**b**). Take DSE1 for instance, the assimilation analysis is performed at the intervals of 3 hours from 11:00 to 17:00 and the rolling forecast is made with a horizon of 24 hours based on the assimilation analysis. The NTEnKF and EnKF is performed in turn.

In all EnKF-based experiments, the ensemble number N is set to 32, which is found to be sufficient to represent the uncertainty in the dust emission field while remaining computationally affordable. Testing with N greater than 32 shows only limited improvements. For NTEnKF experiments, the model ensembles are inevitably expanded as they incorporate ensemble simulations from neighboring instants. To fully cover the potential positions of the dust plume, neighboring times with ± 2 hours apart are empirically chosen in this paper. As demonstrated in Table 1, the ensemble number in DSE1 is extended to 160 when NTEnKF is applied, and the neighboring time stamps of 9:00, 10:00, 12:00, and 13:00 are selected. The 160 ensemble dust simulations are updated according to the EnKF principles and forwarded synchronously for the new rolling forecast; they will serve as the priori in the subsequent assimilation analysis. In practice, the choice of neighboring time ensembles should be able to cover the uncertainty in position when the NTEnKF is applied.





**Table 1.** Experiment configurations (take DSE1 for instance)

| Name | Running ensemble number | Initial assimilation time set | Localization distance (km) |
|------|------|------|------|
| *Control run* | 32 | None | None |
| *EnKF* | 32 | 11:00 | None |
| *L500* | 32 | 11:00 | 500 |
| *NTEnKF* | 160 | 09:00 + 10:00 + 11:00 + 12:00 + 13:00 | None |
| *NTL500* | 160 | 09:00 + 10:00 + 11:00 + 12:00 + 13:00 | 500 |

## 4 Results and discussions

The results are discussed in the aspects of assimilation analysis and model forecast. The benefits of using our NTEnKF algo-
rithm for the dust storm simulation with position errors are emphasized.

### 4.1 Assimilation analysis

Figure 4 displays the spatial distribution of ground BR-PM10 observations (scatter) and dust field forecasts from the average of
the ensembles (panel a.1), the posteriori from EnKF analysis (panel a.2) and EnKF with localization (panel a.3), the average of
the enlarged ensembles (panel b.1), the posteriori from NTEnKF analysis (panel b.2) and NTEnKF analysis with localization
(panel b.3) at 11:00, 15th March, 2021 China Standard Time (CST). It should be noted that the average dust concentrations
in panel b.1 are calculated from the 160 ensemble simulations used in NTEnKF, which slightly differ from the average of 32
ensembles. In DSE1, the RMSE and NMB from the pure ensemble model simulation are as high as 831.5 $\mu$g m$^{-3}$ and -79.8 %.
Both EnKF and LEnKF assimilation analyses achieve very limited improvement in estimating the dust state field. As shown in
panel a.2 and panel a.3, the RMSE and NMB remain high at 830 $\mu$g m$^{-3}$ and -80.8 % in *EnKF*, and 813.4 $\mu$g m$^{-3}$ and -79.9
% in *L500*. The main reason for this is the imbalanced uncertainty between the ensemble simulations and the observations,
as described in Sect. 3.2. As observed in the light blue box in panel a.1, the simulated dust plume is located farther southeast
compared to the PM$_{10}$ measurements. This snapshot exhibits an apparent position error. After EnKF analysis, the simulated
dust plume in the light blue box remains virtually unchanged, as depicted in panel a.2. Numerous ground stations in this area
report high PM$_{10}$ concentrations, but the assimilated dust field fails to resolve any of them. The localization method offers
minimal assistance in this situation, as illustrated in panel a.3. With the unresolved positional error, the EnKF, which focuses
solely on intensity correction, is much less effective.

When it comes to the NTEnKF analysis result, an improved dust field can be noticed. Concerning the Root Mean Square
Error (RMSE) and Normalized Mean Bias (NMB), the two priors depicted in panels a.1 and b.1 exhibit highly similar per-
formances. However, slight differences do exist. For instance, the average of the expanded 160-member ensemble used in
NTEnKF displays a marginally broader spread. The increased ensemble size provides more room for representing background





uncertainties. The enhanced capacity for this is best illustrated in Fig. 6 (a), which exhibits the uncertainty quantified by the enlarged ensemble simulations in NTEnKF formulations. High uncertainty values are seen in pixels where large model-minus-observation errors are present, such as within the light blue box. This allows the posteriori to be adjusted in order to better conform to the observations. In contrast, the relatively low uncertainty over these areas depicted in Fig. 1 (b.2) suggests that

the EnKF method is highly confident in the absence of aerosols and does not require any modification. The observations are effectively assimilated in the NTEnKF analysis. As displayed in panel b.2, the dust plume within the light blue box is adjusted to better match the observations. In particular, the dust to the east of the marked region is well represented in comparison to the posteriori of *EnKF*. The RMSE and NMB are reduced to 755.8 $\mu$g m$^{-3}$ and -76.3 %. Moreover, the posteriori of *NTL500* yields an improved dust field with the RMSE and NMB further reduced to 685.3 $\mu$g m$^{-3}$ and -65.1 %. The implementation

of the localization method eliminates spurious correlations and generates a background error covariance that more accurately describes the model uncertainties. Despite the noticeable improvements achieved in DSE1, the residual errors, as indicated by the RMSE and NMB metrics, remain relatively high. This is primarily due to certain ground stations with extremely high PM$_{10}$ concentrations exceeding 4000 $\mu$g m$^{-3}$ (black scatters). In particular, the western extent of the dust plume is affected by the insufficient number of stations in the area, which results in an inadequate representation of the dust load. Consequently, the

NTEnKF fails to reproduce the dust plume accurately in these regions. Both the model-based and assimilation-based forecasts are unable to reproduce these extreme concentrations, which leads to a partial bias in the metrics. Further evaluation of the dust forecast will be conducted to investigate the further effects of these biases.

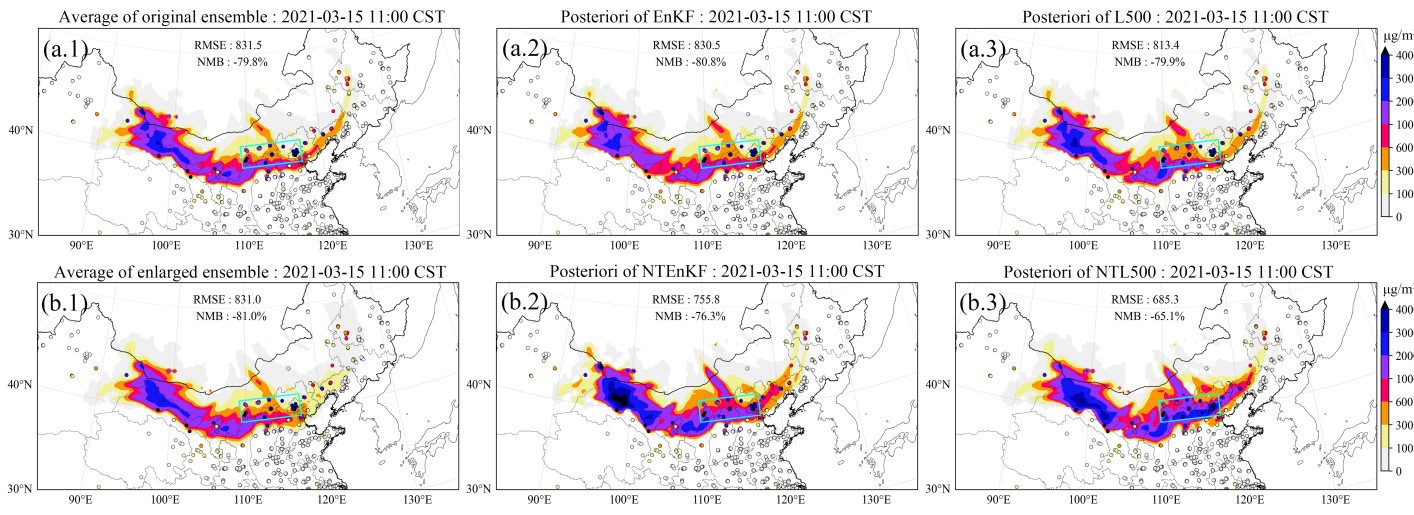

**Figure 4.** Spatial distribution of ground-based BR-PM$_{10}$ observations (scatter) and simulated dust plume (SDP) on surface from central time ensemble model mean (**a.1**), the posteriori SDP updated by EnKF (**a.2**), the posteriori SDP updated by EnKF with localization (**a.3**), central and neighboring time ensemble model mean (**b.1**), the posteriori SDP updated by NTEnKF (**b.2**), the posteriori SDP updated by NTEnKF with localization (**b.3**) at 11:00, 15th March 2021 (CST).





Figure 5 shows the spatial distribution of ground BR-PM$_{10}$ observations (scatter) and dust concentration forecast from ensemble model mean (panel a.1), EnKF (panel a.2) and LEnKF analysis (panel a.3), the average of the enlarged model ensembles

(panel b.1), NTEnKF (panel b.2) and NTEnKF analysis (panel b.3) at 8:00, 28th March, 2021 CST. At this assimilation snapshot in DSE2, the model simulated dust field was found to move farther southeast as shown in panel a.1. As can be seen in light blue box in panel (a.1), the model simulated plume has generally covered the observations both loaded with and free of high-value PM$_{10}$ concentration measurements. Hence, the EnKF analysis is actually still effective in this case. The RMSE and NMB is reduced sharply to 273 $\mu$g m$^{-3}$ and -48.7 % in *EnKF*. They are further reduced to 234.1 $\mu$g m$^{-3}$ and -39.4 % with

the aid of localization method in *L500* case.

The RMSE and NMB of the priori of *NTEnKF* are 391.5 $\mu$g m$^{-3}$ and 26.9 %, which are already much better than performance of the EnKF priori. These improvements can be explained: By averaging these extra temporal ensembles, the ensemble mean of the dust field is reduced and the overestimation is amended. Using NTEnKF assimilation, the RMSE of the posteriori further drops to 202.8 $\mu$g m$^{-3}$ and the NMB is around -33.4 % in *NTEnKF*. These error and bias are much lower through

using the better scaled background covariance shown in Fig. 6 than those obtained in the *EnKF*. What's more, by introducing localization, the RMSE and NMB is further reduced to 184.7 $\mu$g m$^{-3}$ and -29.8 % in *NTL500*. The dust load in the light blue box (panel b.3) is firstly reproduced exactly in its actual range (2000~3000 $\mu$g m$^{-3}$) among all experiment sets.

Figure 5 presents the spatial distribution of ground-based BR-PM$_{10}$ observations (scatter) and dust concentration forecasts from the average of model ensembles (panel a.1), EnKF (panel a.2), and LEnKF analysis (panel a.3), as well as the average

of the enlarged model ensembles (panel b.1), NTEnKF (panel b.2), and NTEnKF analysis (panel b.3) at 8:00, March 28th, 2021 CST. During this assimilation snapshot in DSE2, the model-simulated dust field is observed to have moved further southeast, as depicted in panel a.1. As illustrated by the light blue box in panel a.1, the model-simulated plume generally encompasses observations with and without high-value PM$_{10}$ concentration measurements. Consequently, the EnKF analysis remains effective in this case. The RMSE and NMB are significantly reduced to 273 $\mu$g m$^{-3}$ and -48.7 % in the *EnKF* scenario,

with further reductions to 234.1 $\mu$g m$^{-3}$ and -39.4 % when the localization method is employed in the *L500* case.

The RMSE and NMB of the priori for *NTEnKF* are 391.5 $\mu$g m$^{-3}$ and 26.9 %, which already outperform the EnKF priori. These improvements can be attributed to the averaging of temporal ensembles, which reduces the ensemble mean of the dust field and corrects overestimation from original ensembles. With NTEnKF assimilation, the RMSE of the posterior further decreases to 202.8 $\mu$g m$^{-3}$, and the NMB is approximately -33.4 % in *NTEnKF*. These error and bias values are significantly

lower than those obtained with the *EnKF*, thanks to the better-scaled background covariance displayed in Fig. 6. Moreover, by incorporating localization, the RMSE and NMB are further reduced to 184.7 $\mu$g m$^{-3}$ and -29.8 % in *NTL500*. The dust load within the light blue box (panel b.3) is accurately reproduced within its actual range (2000~3000 $\mu$g m$^{-3}$) for the first time among all experimental sets.

Experiments on DSE3 is also carried out although there is not clear position error captured. Figure S1 presents a snapshot of

ground BR-PM$_{10}$ observations, the priori and the posterior in both the EnKF and NTEnKF assimilation at 20:00, 15th April, 2021 (CST). As shown in panel (a.1), the priori has overestimated the dust field greatly both in spatial coverage and intensity. The RMSE and NMB are as high as 566 $\mu$g m$^{-3}$ and 137.0 %. After EnKF analysis, they are reduced largely to 159.1 $\mu$g m$^{-3}$



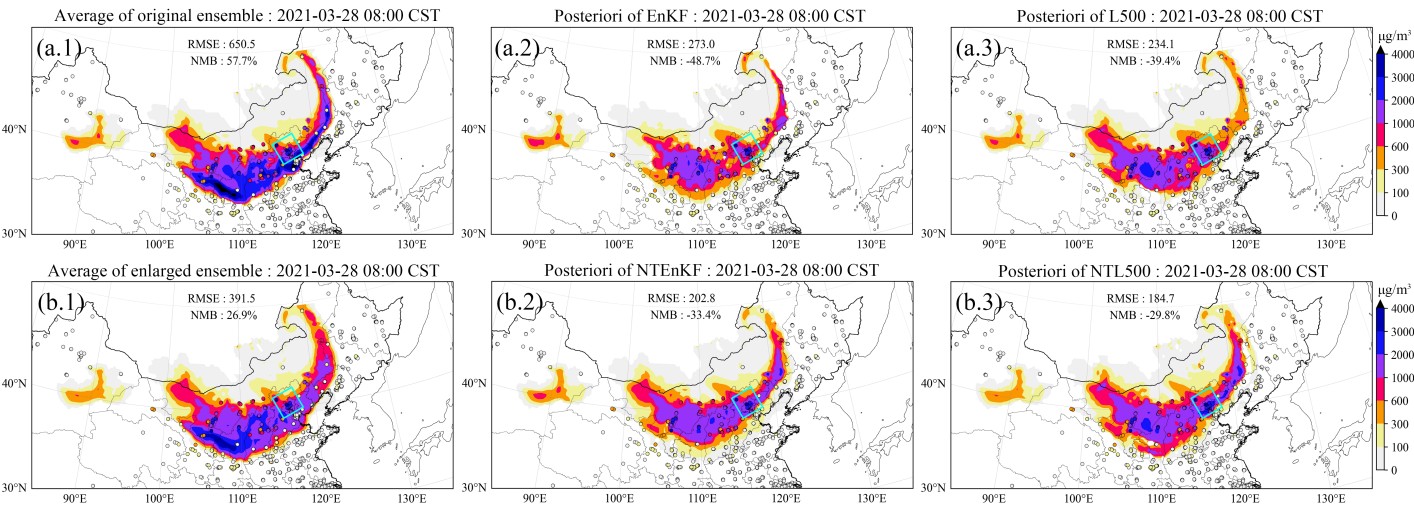

**Figure 5.** Spatial distribution of ground-based BR-PM$_{10}$ observations (scatter) and simulated dust plume (SDP) on surface from central time ensemble model mean (**a.1**), the posteriori SDP updated by EnKF (**a.2**), the posteriori SDP updated by EnKF with localization (**a.3**), central and neighboring time ensemble model mean (**b.1**), the posteriori SDP updated by NTEnKF (**b.2**), the posteriori SDP updated by NTEnKF with localization (**b.3**) at 18:00, 28th March 2021 (CST).

and -28.1 %. The localization helps a little in further improving dust field estimation as the error in the pure *EnKF* is already quite low. When applying the NTEnKF analysis, the RMSE is further reduced to 139.9 $\mu$g m$^{-3}$ and -22.5 %. In the end, 425 *NTL500* shows the lowest RMSE of 126.6 $\mu$g m$^{-3}$ and NMB of -19.9 %. The NTEnKF therefore provides very limit helps to improve the dust field when the simulated dust plume is much less effected by the position error.

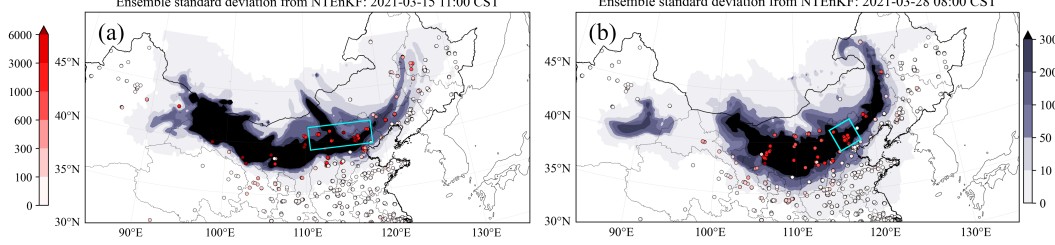

**Figure 6.** Spatial distribution of standard deviation from model ensembles with scatter of model-minus-observation differences (absolute value) at 11:00 in DSE1(**a**) and 08:00 in DSE2(**b**). The initial assimilation analysis is performed at these time. Colorbar left is for model-minus-observation differences and right is for standard deviation.



## 4.2 Forecast skills

The posteriori of the 3 dimensional dust field was then used as the initial condition for new rolling forecast. Figure 7 shows the assimilation-based 3-hour forecast starting from initial assimilation analysis at 11:00 and second assimilation analysis at
14:00 15th March 2021, respectively. Two experiment results, *L500* and *NTL500*, are selected to illustrate the adding-value of our NTEnKF.

As depicted in Fig. 7 (a), the forecast from the *L500* scenario at 14:00, following the initial assimilation analysis at 11:00, remains inconsistent with the observations. The forecasted dust plume still exhibits discrepancies. The improvement over the pure model forecast shown in Fig. 1 (a.3) is relatively limited. The dust concentration continues to be substantially underesti-
mated due to the persistent position error, with the NMB remaining steady at approximately -80 %. By employing the NTEnKF assimilation, an enhanced dust plume forecast is obtained, as illustrated in panel b. The heterogeneous underestimation of the prior model is now adjusted to better align with the observations. Furthermore, the position error in the *L500* forecast is recti-fied. The improvements brought about by the NTEnKF assimilation are particularly evident for the dust within the light blue box. The forecast from the *L500* case suggests that the area is less affected by dust, whereas the ground $PM_{10}$ measurements
indicate dust concentrations as high as 2000 $\mu$g m$^{-3}$. In contrast, *NTL500* delivers the most accurate dust forecast. Although some underestimation persists to the east of the box, the dust distribution is more thoroughly reproduced, with the bias reduced to -48 % at 14:00 and -42.1 % at 17:00.

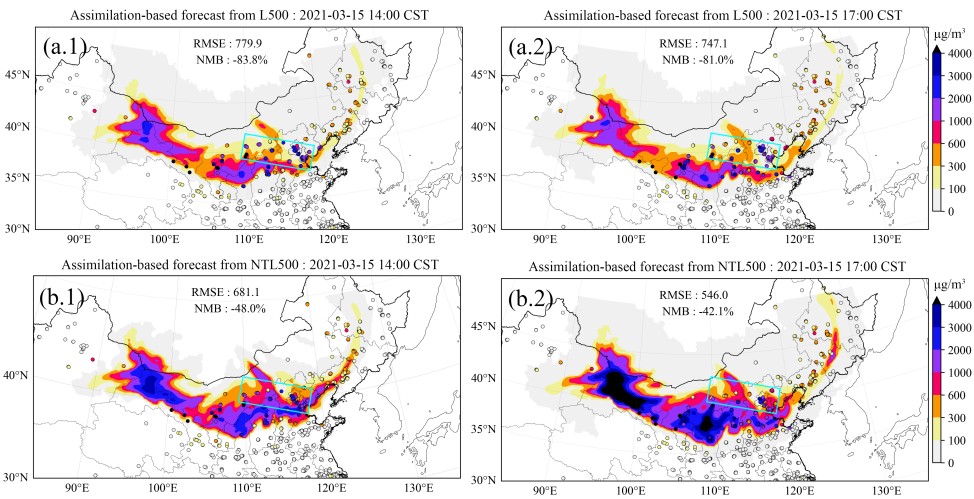

**Figure 7.** Spatial distribution of ground-based BR-$PM_{10}$ observations (scatter) and assimilation-based dust plume forecast on surface starting from the previous assimilation analysis at 11:00 (left) and 14:00 (right) 15th March, 2021 CST. Figures up are from *L500* at 14:00 (**a.1**) and 17:00 (**a.2**). Figures below are from *NTL500* at 14:00 (**b.1**) and 17:00 (**b.2**).

Figure 8 displays two assimilation-based forecast scenarios in DSE2. Similar to DSE1, the EnKF-based forecast in DSE2 also underestimates the actual dust intensity heterogeneously. The dust concentration north of the forecasted dust is consider-



ably lower than the observations, indicating that the two EnKF assimilation analyses do not fully correct this position error. The NMB remains high at -57.8 % at 11:00 and -63.9 % at 14:00. Upon applying NTEnKF, the position error is corrected to a certain extent. The overall concentration of the 3-hour dust forecast increases, which better fits to the observations. The NMB is reduced to -28.1 % at 11:00 and -26.1 % at 14:00. However, this improvement is not reflected in the RMSE at 11:00, as the northeastern extent of the dust is inconsistent with the observations. This inconsistency primarily arises due to the lack of ground stations in the area at the time of the initial assimilation analysis at 8:00. After the second assimilation analysis, this issue is mitigated, and the RMSE is reduced to 244.7 $\mu$g m$^{-3}$.

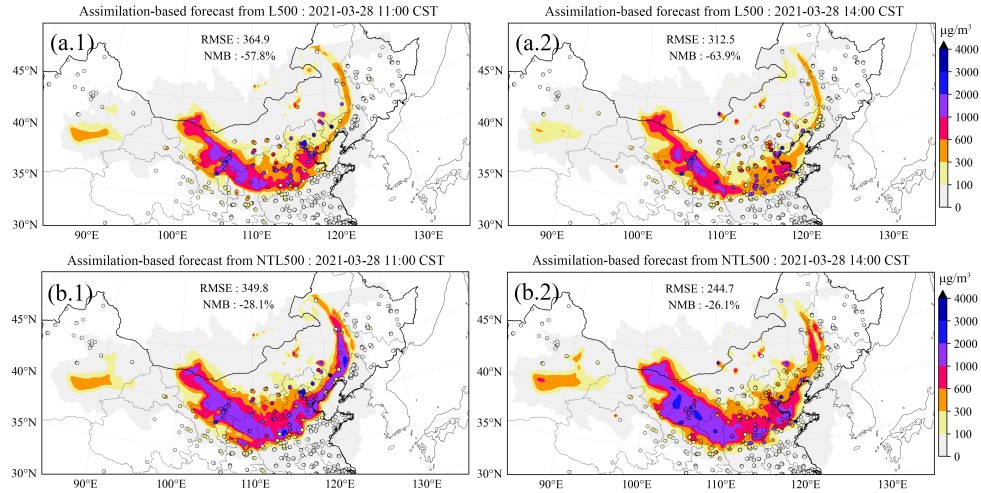

**Figure 8.** Spatial distribution of ground-based BR-PM$_{10}$ observations (scatter) and assimilation-based dust plume forecast on surface starting from the previous assimilation analysis at 08:00 (left) and 11:00 (right) 28th March, 2021 CST. Figures up are from *L500* at 11:00 (**a.1**) and 14:00 (**a.2**). Figures below are from *NTL500* at 11:00 (**b.1**) and 14:00 (**b.2**).

In addition to the snapshots of the dust forecast, an comprehensive evaluation of forecast skills is also necessary to see the performance of NTEnKF algorithm. A general evaluation on the forecasting skills is carried out in this section.

Figure 9 presents the time series of RMSE and NMB for the 24-hour dust forecast after three assimilation analyses in DSE1 (starting from 11:00, 14:00, and 17:00). In these cases, the *Control run* generates a dust field characterized by a high RMSE (ranging from over 800 $\mu$g m$^{-3}$ to around 600 $\mu$g m$^{-3}$) and a large NMB (consistently around -85 %). The EnKF analysis, however, does not improve this dust forecast. In fact, the RMSE and NMB of the dust forecast from the *EnKF* scenario are nearly identical to the *Control run*, as evidenced by the comparison between the black dashed line and the blue line in panel a. This result can be primarily attributed to the position error discussed in Sect. 2.4. The EnKF algorithm offers minimal assistance in correcting the model simulation when position errors are present. These errors are not occasional but cumulative, as demonstrated in the subsequent two assimilation timestamps at 14:00 and 17:00, during which the assimilation analysis fails to improve the situation. Moreover, it has been observed that the localization method is unable to enhance the forecast in the





presence of position errors. Similar for NMB, as depicted in panel b, the improvements are also insignificant. The NMB for the *Control*, *EnKF*, and *L500* scenarios remains consistently around -85 % throughout the entire forecast time range.

By applying the NTEnKF analysis, a clear reduction of RMSE is observed in panel a. There is an approximate decrease of 100 $\mu$g m$^{-3}$ in *NTEnKF* compared to *EnKF*, which indicates that the NTEnKF analysis effectively corrects the position error. At the subsequent assimilation timestamps, this situation improves, with an even greater decrease in RMSE. The RMSE of *NTL500* is slightly lower than that of *NTEnKF*, except for the time after 3:00 on 16th March during the third assimilation timestamp. As for NMB, quite promising results are achieved. In *NTEnKF*, the NMB decreases stepwise at three time points, 470    from around -70 % at 11:00 to around -50 % at 14:00, and finally to around -40 %. The NTEnKF algorithm gradually takes effect over the three assimilation analyses. In *NTL500*, the localization method demonstrates its efficacy, especially after the third assimilation timestamp at 17:00. The NMB is reduced to around -20 %, which is significantly lower than that of the *L500*.

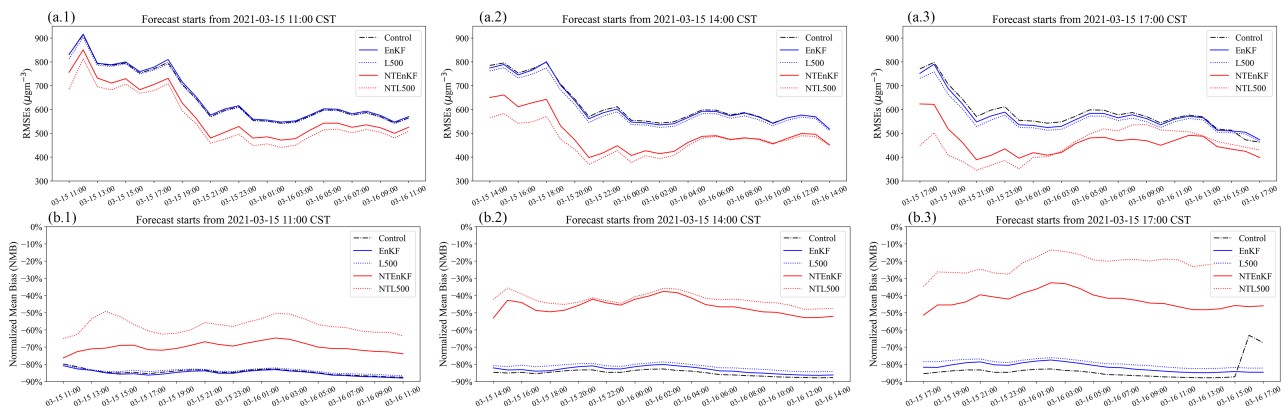

**Figure 9.** Time series of 24-hour Root Mean Square Error (RMSE) on the dust forecast starting from 11:00 (**a.1**), 14:00 (**a.2**), 17:00 (**a.3**) and normalized mean bias (NMB) starting from 11:00 (**b.1**), 14:00 (**b.2**), 17:00 (**b.3**) on 15th March 2021.

Figure 10 displays the time series of RMSE and NMB on a 24-hour dust forecast after three assimilation analyses in DSE2. It is important to note that a nonlinear y-axis is used in DSE2 because the metrics of *Control run* significantly differ from all the 475    assimilation-based forecasts. Unlike DSE1, *EnKF* in DSE2 does improve the dust forecast in terms of RMSE and NMB. The RMSE drops from around 700 $\mu$g m$^{-3}$ to less than 400 $\mu$g m$^{-3}$ at the initial assimilation timestamp (8:00). Overestimation of the dust load is also alleviated, as shown in panel b.1. However, no further reduction is observed at subsequent time points. As can be seen in panels a.2 and a.3, the RMSE of *EnKF* remains almost constant compared to panel a.1. This indicates that the position error is not corrected, and it constitutes part of the RMSE that is difficult to eliminate. The trend of NMB also reflects 480    this situation. *L500* is unable to correct the position error, although it does help reduce the error to some extent.

   In the scenario of the NTEnKF analysis, an improvement in the dust forecast of DSE2 is obtained. A general reduction of RMSE (around 50 $\mu$g m$^{-3}$) in *NTEnKF* compared to *EnKF* can be seen in panel a.1. Furthermore, in the subsequent forecasts, a steady decrease in RMSE is noted. The RMSE fluctuates around 250 $\mu$g m$^{-3}$ after 11:00 and 200 $\mu$g m$^{-3}$. *NTL500* exhibits





a similar pattern to *NTEnKF* for most of the forecast. Considering the NMB, their differences become apparent. As shown in
panel b, the NMB of *NTL500* clearly demonstrates its superiority over *NTEnKF*, with a maximum reduction of 10 %. In DSE2,
the *EnKF* and *L500* have already achieved well-reproduced dust fields, while the *NTEnKF* and *NTL500* can further improve
these fields by correcting the position error.

24-hour forecast based on the assimilation analysis is also made on DSE3 as shown in Fig. S2. In DSE3, the NTEnKF
analysis helps to reduce the RMSE and NMB most of the time except few hours after the initial assimilation time. The RMSE
after assimilation analysis at 20:00 rises over the *EnKF* and returns to a low level at the following assimilation time stamp. This
is mainly because the time consistency is damaged to some degree. It can be made up after few hours of forecast. In this case
which shows no position error, NTEnKF can also improve the dust forecast. Meanwhile, a more flexible choice of neighboring
time and localization in time sets can be adopted to alleviate the damage to time consistency and reduce computational cost.

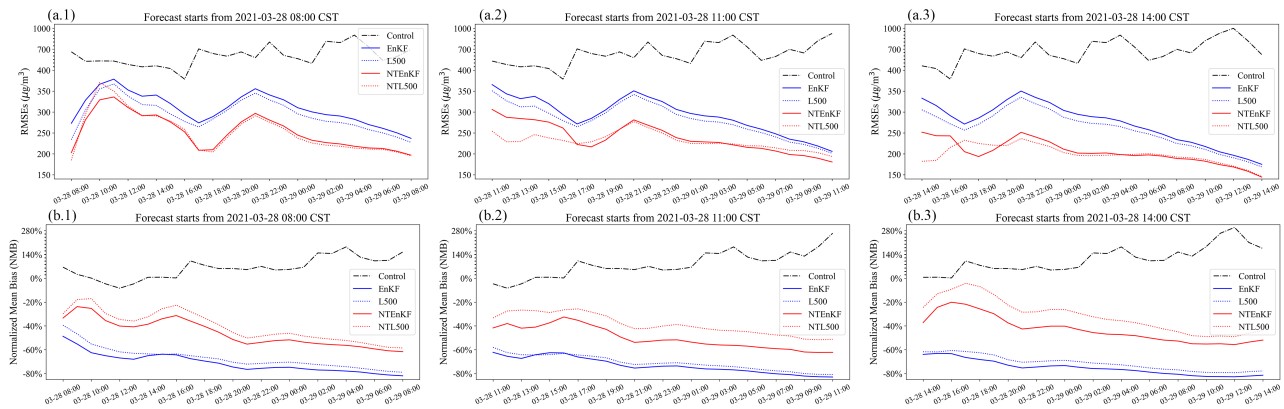

**Figure 10.** Time series of 24-hour Root Mean Square Error (RMSE) on the dust forecast starting from 08:00 (**a.1**), 11:00 (**a.2**), 14:00 (**a.3**)
and normalized mean bias (NMB) starting from 08:00 (**b.1**), 11:00 (**b.2**), 15:00 (**b.3**) on 28th March 2021.

## 5   conclusions

The Chemistry Transport Model (CTM) is a powerful tool for forecasting air pollutants. However, as a simplified version
of the real atmospheric world, it suffers from various deficiencies, particularly in two major uncertainties: emissions and
meteorology. Uncertainty from meteorological fields can cause model forecast errors, especially in long-distance transport. In
dust storm forecasting applications, a position error is noted that significantly degrades the overall performance of the forecast
and prevents the EnKF assimilation algorithm from effectively incorporating observational data.

The EnKF primarily focuses on intensity adjustment, and its background error covariance is generally designed to represent
the uncertainty of intensity characteristics in model simulations. However, when a position error exists between the simulation
and observations, such as the long-distance dust storm tracking in this study, the EnKF is incapable of thoroughly resolving
the observations. Observations over low model uncertainty pixels are 'ignored' by the EnKF algorithm. To address this issue, a



neighboring time ensemble Kalman filter (NTEnKF) is proposed. NTEnKF introduces uncertainty of the dust plume position
into the background error covariance by incorporating extra ensemble simulations at neighboring time instances. This enlarged
ensemble not only reflects the uncertainty of dust intensity but also reveals the potential positions of the plume, allowing for
more accurate and effective assimilation and improving dust storm forecasting.

The NTEnKF algorithm was tested on three super dust storm events (DSE1, DSE2, and DSE3) that occurred in Spring 2021.
Several experiments were designed to examine the performance of the NTEnKF algorithm in these cases, with a focus on
differences between EnKF and NTEnKF. In terms of assimilation analysis, the NTEnKF analysis corrected the position error
in DSE1 to a large extent. Comparison between the standard deviations from posteriori of EnKF and NTEnKF explained for
it. The standard deviations from NTEnKF analysis indicated wilder potential dust spread and were more consistent with the
model-minus-observation. Observations that were 'ignored' by EnKF were comprehensively resolved in NTEnKF, resulting in
decreased RMSE and NMB. For DSE2, the position error was not as significant as in DSE1; however, imbalanced uncertainties
were also observed. Nevertheless, NTEnKF still produced an improved dust field with lower RMSE and NMB compared to
EnKF. In both cases, the localization method helped reduce RMSE and NMB.

Regarding the forecast performance, promising results were obtained. In DSE1, the RMSE and NMB revealed that EnKF
barely resolved the observations, resulting in a dust forecast similar to the pure model results. In contrast, NTEnKF provided
a dust field forecast with reduced errors, especially in terms of NMB. Snapshots of following forecasts and trend of metrics
proved the superiority of NTEnKF. In DSE2, an improved dust field forecast was also observed even though EnKF analy-
sis already provided a dust field with relatively low RMSE and NMB. Additionally, the localization method contributed to
further reducing the error. Overall, the NTEnKF algorithm demonstrated improved performance in assimilation analysis and
forecasting for the tested dust storm events compared to the traditional EnKF approach.

*Code and data availability.* The NTEnKF code is archived on Zenodo at https://doi.org/10.5281/zenodo.7611976 (Pang, 2023). The $PM_{10}$
data used in this study is also archived on Zenodo at https://doi.org/10.5281/zenodo.6459866 (Jin, 2022). The real-time $PM_{10}$ data established
by the Ministry of Ecology and Environment is available to the public at https://quotsoft.net/air (Wang, last access: August 2023). The source
code and user guide of the LOTOS-EUROS model could be obtained from https://lotos-euros.tno.nl (TNO, last access: August 2023).

*Author contributions.* JJ conceived the study and designed NTEnKF algorithm. MP wrote the code of the assimilation and carried out the
prediction and evaluation. AS, WH, JX, LF, JL, HXL and HL provided useful comments on the paper. MP and JJ prepared the manuscript
with contributions from HJ and all others co-authors.

*Acknowledgements.* This work is supported by the National Natural Science Foundation of China [grant No. 42105109 and 42205031] and
Natural Science Foundation of Jiangsu Province (NO. BK20210664).





**Competing interests**

The authors declare that they have no conflict of interest.





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
