# Peer review of "Valid time shifting Ensemble Kalman filter (VTS-EnKF) for dust storm forecasting"

_Geoscientific Model Development, 2023_

## Author Comment (AC1)

**Authors' Response to Reviews of**

**Valid time shifting Ensemble Kalman filter (VTS-EnKF) for dust storm forecasting**

Mijie Pang, Jianbing Jin*, Arjo Segers, et al.
*Geoscientific Model Development Discussions,* `10.5194/gmd-2023-219`
* * *
**RC:** *Reviewers' Comment*,     AR: Authors' Response,     ☐ Manuscript Text

**1. Overview**

Response to Referee #1: We would like to thank the referee for the careful review throughout the paper and the in-depth comments that help to improve our paper.

**2. Grave concerns**

**RC:**  *Grave concern 1: Contrary to the authors' claims, the Neighbouring Time approach to aggregating ensemble members is not novel.*

*The authors claim that their neighbouring time approach to increase ensemble sizes is novel. Unless I am missing some detail in their manuscript, that claim is incorrect. In fact, that method has been extensively tested. The common name for that method is "valid time shifting" (VTS). A similar, but extremely popular, variant of this method is "time-lagged ensembles". This concern is particularly grave for this study because this ensemble size increasing method is a huge part of this study's supposed novelty.*

*Here is a sampling of papers that employed those methods.*

*Gasperoni, N. A., X. Wang, and Y. Wang, 2023: Valid Time Shifting for an Experimental RRFS Convection-Allowing EnVar Data Assimilation and Forecast System: Description and Systematic Evaluation in Real Time. Mon. Wea. Rev., 151, 1229–1245, https://doi.org/10.1175/MWR-D-22-0089.1.*

*Huang, B., and X. Wang, 2018: On the Use of Cost-Effective Valid-Time-Shifting (VTS) Method to Increase Ensemble Size in the GFS Hybrid 4DEnVar System. Mon. Wea. Rev., 146, 2973–2998, https://doi.org/10.1175/MWR-D-18-0009.1.*

*Xu, Q., L. Wei, H. Lu, C. Qiu, and Q. Zhao, 2008: Time-expanded sampling for ensemble-based filters: Assimilation experiments with a shallow-water equation model. J. Geophys. Res., 113, D02114, https://doi.org/10.1029/2007JG000450.*

*Van den Dool, H. M., and L. Rukhovets, 1994: On the weights for an ensemble-averaged 6–10-day forecast. Wea. Forecasting, 9, 457–465, https://doi.org/10.1175/1520-0434(1994)009<0457:OTWFAE>2.0.CO;2.*

*To address this concern, please remove all claims that the Neighboring Time approach is novel in your manuscript.*

 AR:  Thanks for the comment. We have removed all the improper statement in the paper. The corresponding papers

are cited. Details are below:

> In this paper, the standard EnKF assimilation is coupled with a valid time shifting (VTS) method (Xu et al., 2008; Lu et al., 2011; Zhao et al., 2015; Huang and Wang, 2018) for better resolving the position error in long-distance dust storm transport simulation. This assimilation methodology is referred to as VTS-EnKF throughout this paper. For assimilation analysis at a given time, the background error covariance of the simulated dust plume is calculated using not only the original ensemble simulation, but also the same ensemble simulations at neighboring moments (a few hours earlier and later) (Gasperoni et al., 2022, 2023). These extra ensemble members represent the potential position spread of the actual dust plume, effectively accounting for transport errors. The resampled ensemble members quantify the complex covariance that captures both intensity and position error dynamics, without requiring additional processing on observations, meteorological fields, or other physical parameters. We tested the VTS-EnKF on two severe dust storm events that occurred in 2021. Our results show superior assimilation performance compared to the standard EnKF, particularly when position errors are present in the simulated dust plume.

**RC:** *Grave concern 2: Their EnKF's struggle with positioning error is highly contrived.*

*The EnKF's struggle with positioning errors is likely simply due to their choice of meteorological forcing. Specifically, they failed to account for uncertainties in meteorological forcing. This could have been avoided by using the ECMWF's ensemble forecasts instead of the operational forecast. With an ensemble of forecasts, there should be more ensemble spread in the positioning of the dust storms, thus ameliorating the EnKF's issue with positioning error.*

*To address this concern, the authors need to rerun all of their experiments using the ECMWF's ensemble forecast data. This will likely take months of effort. The ECMWF has archived some of its ensemble forecasts on MARS. The ERA5's 10-member ensemble is also available through the Climate Data Store.*

AR: Thanks for the in-depth comment. Now we have re-design our assimilation system: Each of our ensemble simulation (N=32) is driven by the perturbed emission and perturbed meteorology input. European Center for Medium-ranged Weather Forecast (ECMWF) ensemble forecast (totally 51 ensembles) are used, and 32 of them are randomly picked out for driving our ensemble simulation. It turns out that only trivial wilder ensemble spread found and the results are quite similar to the original EnKF experiments. We believe that the position error is mainly caused by the meteorology while ECMWF's ensemble forecast is not sufficient to account for the position error. By our method, the error can be alleviated.

Below are the new descriptions about the consideration of both emission and meteorology uncertainties:

[revised manuscript text omitted]

**RC:** *Grave concern 3: The authors did not satisfactorily demonstrate that the NTEnKF's improved performance over EnKF is purely due to NTEnKF's ability to handle positioning errors. The fact that the NTEnKF has less sampling error-related under-dispersion is likely playing a role.*

*To address this concern, run an experiment with the NTEnKF that uses the same number of members as*

[Figure]

**Figure 6.** Spatial distribution of standard deviation from model ensembles with scatter of model-minus-observation differences (absolute value) at 11:00 in DSE1(**a**) and 08:00 in DSE2(**b**). The initial assimilation analysis is performed at these time. Colorbar left is for model-minus-observation differences and right is for standard deviation.

*the EnKF (i.e., run a 32-member NTEnKF experiment). I suspect that the NTEnKF's performance will be comparable to the EnKF's in such a situation. Remember to use the ECMWF ensemble forecasts as your meteorological forcings.*

AR:   Agree with the referee. We now run experiments with the VTS-EnKF with same ensemble number as EnKF, which is 32. We separated the original 32 ensemble into [6,6,8,6,6] and choose the neighbouring time following the original time set (-2, -1, 0, +1, +2 hour). We found that by this limited ensemble number, VTS-EnKF provides slightly better performance than the EnKF. The sampling error indeed plays a role under the smaller ensembles. However, by applying the localization, this error can be noticeably reduced. It proves that our VTS-EnKF's superiority in handling the position error.
* * *
**4.2 Assessment of smaller ensembles**

To further assess the performance of VTS-EnKF, VTS-EnKF experiments with same ensembles as the EnKF are designed. They are referred to as *VTS-EnKF-small* and *VTS-L500-small*, respectively. The total 32 ensembles are composed of 8 central ensembles and 4×6 ensembles from neighboring ±1 and ±2 hours. Figure 9 displays the time series of RMSE and NMB on a 24-hour dust forecast after three assimilation analyses in DSE1. In terms of RMSE, *VTS-EnKF-small* only shows slightly better performance than the EnKF. This mostly caused by the sampling error arises from limited ensembles resampled from the central ensembles (only 8 ensembles). However, by applying the localization, the RMSE is noticeably reduced by 100 µg m$^{-3}$. The performance is comparable to the *VTS-L500* (red dash line) with totally 160 ensembles. By mitigating the sampling error, the VTS-EnKF's capability of handling the position error can be revealed, which can be noticed by comparison with *L500* and *VTS-L500-small*. This improvement can be better seen in NMB. NMB of *VTS-L500-small* is much lower than the *EnKF* and *L500*. Its performance is also comparable to the *VTS-L500* with 160 ensembles.

Same experiments on DSE2 are also carried out. Results can be found in Fig. S2 in supporting information. Similar to DSE1, the *VTS-EnKF-small* achieves slightly better RMSE and NMB than *EnKF* and *L500*. While in *VTS-L500-small*, noticeable improvements can been found especially for the forecast after the second and last assimilation. Reduction of 100 µg m$^{-3}$ in RMSE and 20% in NMB are obtained.

[Figure]

**Figure 9.** Time series of 24-hour Root Mean Square Error (RMSE) on the dust forecast starting from 11:00 (**a.1**), 14:00 (**a.2**), 17:00 (**a.3**) and normalized mean bias (NMB) starting from 11:00 (**b.1**), 14:00 (**b.2**), 17:00 (**b.3**) on 15th March 2021.

[Figure]

**Figure S2.** Time series of 24-hour Root Mean Square Error (RMSE) on the dust forecast starting from 11:00 (**a.1**), 14:00 (**a.2**), 17:00 (**a.3**) and normalized mean bias (NMB) starting from 11:00 (**b.1**), 14:00 (**b.2**), 17:00 (**b.3**) on 28th March 2021.

On the other hand, we have also tested the standard EnKF with more ensemble members, but very limited improvements were obtained.

> In EnKF-based experiments, *EnKF* and *L500*, the ensemble number N is set to 32, which is found to be sufficient to represent the uncertainty in the dust simulation while remaining computationally affordable. Testing with N greater than 32 shows only limited improvements.

**3. Major concern**

**RC:** *The authors did not explore the statistical problems that surround the use of Neighbourhood time ensembles. The primary issues are*

*the ensemble members are correlated with each other (i.e., the ensemble is no longer i.i.d.), causing the estimated ensemble variance to be biased from the true forecast variance, and, the ensemble becomes non-Gaussian, especially if time points far apart are used, strengthening the possibility that the EnKF creates suboptimal biased analyses*

**AR:** The choice of neighbouring time ensembles definitely affects the analysis. An improper choice of interval can lead to undesirable analysis, such as less-effective ensemble members (interval too small) or unrepresentative ensemble covariances (interval too large). We didn't explore the choice of neighbouring time interval in our original manuscript. Here, we design new experiments that use different time intervals to tell the impact of the choice of neighbouring time. Below are the new experiment settings and results:

**Table 1.** Experiment settings.

| Name | Running ensemble number | Initial assimilation time set (hour) | Ensemble set | Localization distance (km) |
|---|---|---|---|---|
| *Control* | 32 | None | [32] | None |
| *EnKF* | 32 | $t$ | [32] | None |
| *L500* | 32 | $t$ | [32] | 500 |
| *VTS-EnKF* | 160 | $t-2, t-1, t, t+1, t+2$ | [32,32,32,32,32] | None |
| *VTS-L500* | 160 | $t-2, t-1, t, t+1, t+2$ | [32,32,32,32,32] | 500 |
| *VTS-EnKF-small* | 32 | $t-2, t-1, t, t+1, t+2$ | [6,6,8,6,6] | None |
| *VTS-L500-small* | 32 | $t-2, t-1, t, t+1, t+2$ | [6,6,8,6,6] | 500 |
| *VTS-EnKF-t1* | 96 | $t-1, t, t+1$ | [32,32,32] | None |
| *VTS-EnKF-t2* | 96 | $t-2, t, t+2$ | [32,32,32] | None |
| *VTS-EnKF-t3* | 96 | $t-3, t, t+3$ | [32,32,32] | None |
| *VTS-EnKF-t4* | 96 | $t-4, t, t+4$ | [32,32,32] | None |
| *VTS-EnKF-t5* | 96 | $t-5, t, t+5$ | [32,32,32] | None |
| *VTS-EnKF-t6* | 96 | $t-6, t, t+6$ | [32,32,32] | None |

**4.4 Sensitivity of time interval**

Previous researches have found that an improper neighboring time interval $\tau$ can lead to undesirable results, such as less-effective ensemble members (interval too small) ($\tau$ too small) or ensemble member clustering and unrepresentative ensemble covariances ($\tau$ too large) (Xu et al., 2008; Gasperoni et al., 2022, 2023). To explore the sensitivity of the choice of neighboring time interval, series of VTS-EnKF experiments with different neighboring time interval were carries out. Time intervals ranging from 1 to 6 hour were tested. As shown in Fig. 10, snapshots from 6 experiments on DSE1 clearly depicts the trend. In general, all the VTS-EnKF experiments show better performance than EnKF. While in terms of specific time interval, different patterns can be noticed. For short intervals including 1 and 2 hour, there is not sufficient ensemble spread to account for the position error. Thus there are still position error remaining and RMSE is still high. For long intervals including 5 and 6 hour, dust plume is clustered away from central dust plume. Three dust branches are noticed in *VTS-EnKF-t5* and an overly backwards dust plume is noticed in *VTS-EnKF-t6*. In this case, 3-hour interval is the best choice with the lowest RMSE (696.11 µg m$^{-3}$) and NMB (-63.5 %).

Same experiments on DSE2 are also performed and snapshots are shown in Fig. S3. Similar patterns are found on DSE2. Lowest RMSE and NMB are achieved in *VTS-EnKF-t4*. Too short interval leads to inability in position error correction and too long interval leads to excessive dust plume. Considering both cases, 3-hour interval is the preferred choice which holds the capability to handle position and not creates excessive clustered dust plume.

[Figure]

**Figure 10.** Spatial distribution of ground-based BR-PM$_{10}$ observations (scatter) and simulated dust plume (SDP) on surface from the posteriori SDP updated by VTS-EnKF-t1 (**a**), the posteriori SDP updated by VTS-EnKF-t2 (**b**), the posteriori SDP updated by VTS-EnKF-t3 (**c**), the posteriori SDP updated by VTS-EnKF-t4 (**d**), the posteriori SDP updated by VTS-EnKF-t5 (**e**), the posteriori SDP updated by VTS-EnKF-t6 (**f**) at 11:00, 15th March 2021 (CST).

[Figure]

**Figure S3.** Spatial distribution of ground-based BR-PM$_{10}$ observations (scatter) and simulated dust plume (SDP) on surface from the posteriori SDP updated by VTS-EnKF-t1 (**a**), the posteriori SDP updated by VTS-EnKF-t2 (**b**), the posteriori SDP updated by VTS-EnKF-t3 (**c**), the posteriori SDP updated by VTS-EnKF-t4 (**d**), the posteriori SDP updated by VTS-EnKF-t5 (**e**), the posteriori SDP updated by VTS-EnKF-t6 (**f**) at 11:00, 28th March 2021 (CST).

**4. Minor comments**

**RC:** *1) The authors' writing seem to imply that the Pf matrix does not normally account for position errors. That is incorrect. The Pf matrix accounts for both intensity and position uncertainties if the forecast ensemble has both kinds of uncertainty. However, note that the Pf matrix only adequately represents position uncertainties if it is sufficiently small – position uncertainties result in non-Gaussian statistics if those uncertainties are large.*

**AR:** Thanks for the comment. We didn't clearly explain the constitutions of the $\mathbf{P}^f$. We agree that the $\mathbf{P}^f$ accounts for both intensity and position errors. While when there are significant position errors, non-Gaussian statistics can be aggravated. Which will mislead EnKF that relies on Gaussian distribution of errors. Descriptions are made in Line 490-493. Details are below:

> The background error covariance of EnKF is generally designed to represent the intensity and position uncertainty. However, when the position error is sufficiently large, the background error covariance can't adequately represent the position error, which is highly non-Gaussian. In the case of the long-distance dust storm tracking, the EnKF is incapable of thoroughly resolving the observations. Observations over low model uncertainty pixels are 'ignored' by the EnKF algorithm.

**RC:** *2) Line 64: Please acknowledge that the EnKF is suboptimal for non-Gaussian problems. Though the EnKF can be employed in such situations, the EnKF is probably injecting some kind of bias because it is designed specifically for Gaussian problems.*

**AR:** Thanks for your comment. We agree that EnKF is Gaussian-dependent. It is added in Line 66-68.

> Meanwhile, inherited from Kalman filter, EnKF relies on Gaussian distribution of error statistics (Amezcua and Van Leeuwen, 2014). For non-Gaussian problems, EnKF can create suboptimal results (Lei et al., 2010).

**RC:** *3) Given the centrality of the EnKF to the paper, it seems unusual that only 3 papers are cited between lines 60-67. In particular, the sentences in lines 66 and 67 are missing supporting references. Here's a good review paper about the EnKF that you can use to find more EnKF references: Houtekamer, P. L., and F. Zhang, 2016: Review of the Ensemble Kalman Filter for Atmospheric Data Assimilation. Mon. Wea. Rev., 144, 4489–4532, https://doi.org/10.1175/MWR-D-15-0440.1.*

*Also, the stochastic EnKF scheme you are using is not the one that Geir Evensen formulated. It is the one Burgers formulated. Here's the paper: Burgers, G., P. Jan van Leeuwen, and G. Evensen, 1998: Analysis Scheme in the Ensemble Kalman Filter. Mon. Wea. Rev., 126, 1719–1724, https://doi.org/10.1175/1520-0493(1998)126<1719:ASITEK>2.0.CO;2.*

*Peter Jan van Leeuwen of Colorado State University (Evensen's good friend), recently published a much more satisfactory explanation of the stochastic EnKF than Burgers et al (1998): van Leeuwen PJ. A consistent interpretation of the stochastic version of the Ensemble Kalman Filter. QJR Meteorol Soc. 2020; 146: 2815–2825. https://doi.org/10.1002/qj.3819*

**AR:** Thanks for these strong supporting reference. We have added more references when describing the EnKF in Line 58-68:

> Filtering methods, on the other hand, assimilate observations sequentially and are more efficient for operational forecasting systems. Various filtering approaches, such as Kalman Filter (Kalman, 1960), Extended Kalman Filter (Brunner et al., 2012), and Particle Filter (Leeuwen et al., 2019), have been developed. Among all the filtering methods, the Ensemble Kalman Filter (EnKF) is the most popular filtering method due to its ability to handle high-dimensional models, easy parallelization (Evensen, 1994; Katzfuss et al., 2016; Houtekamer and Zhang, 2016). It uses limited ensembles to estimate the background error covariance statistics of the model (Hamill, 2006; Houtekamer et al., 2014). Its advantages include handling non-linearity, not requiring explicit calculation of tangent linear operators, and computational efficiency (Bannister, 2017). EnKF has been successfully applied in various disciplines, e.g.,weather forecasting (Houtekamer et al., 2005) and hydrology (Reichle et al., 2002). Meanwhile, inherited from Kalman filter, EnKF relies on Gaussian distribution of error statistics (Amezcua and Van Leeuwen, 2014). For non-Gaussian problems, EnKF can create suboptimal results (Lei et al., 2010).

As to the Peter Jan van Leeuwen's new explanation of the stochastic EnKF, we are quite interested in this interpretation and will examine the impact in the future. Thanks for the recommendation!

**RC:** *4) Eq. 9 – The notation can be mistaken as summing up matrices containing the ensemble members at different time points. Please find another way to mathematically express the idea that you are concatenating ensembles across time. Perhaps you can refer to the valid time shifting papers that I referenced earlier.*

AR: Thanks for the comment. We have used better expression of the idea. Details are below:

[revised manuscript text omitted]

---

## Author Comment (AC2)

**Authors' Response to Reviews of**

**Valid time shifting Ensemble Kalman filter (VTS-EnKF) for dust storm forecasting**

Mijie Pang, Jianbing Jin*, Arjo Segers, et al.
*Geoscientific Model Development Discussions,* `10.5194/gmd-2023-219`
* * *
**RC:** *Reviewers' Comment*,     AR: Authors' Response,     ☐ Manuscript Text

**1. Overview**

Response to Referee #2: We would like to thank the referee for the careful review throughout the paper and the in-depth comments that help to improve our paper.

**2. Comments**

**RC:**   *1) As the other referee pointed out, similar techniques have been proposed before as the "valid time shifting" in other applications. The authors should reconsider their novelty about the method and refer to the proper papers.*

 AR:  Thanks for the comment. We have removed all the improper statement in the paper. The corresponding papers are cited. Details are below: In Line 114-124:

> In this paper, the standard EnKF assimilation is coupled with a valid time shifting (VTS) method (Xu et al., 2008; Lu et al., 2011; Zhao et al., 2015; Huang and Wang, 2018) for better resolving the position error in long-distance dust storm transport simulation. This assimilation methodology is referred to as VTS-EnKF throughout this paper. For assimilation analysis at a given time, the background error covariance of the simulated dust plume is calculated using not only the original ensemble simulation, but also the same ensemble simulations at neighboring moments (a few hours earlier and later) (Gasperoni et al., 2022, 2023). These extra ensemble members represent the potential position spread of the actual dust plume, effectively accounting for transport errors. The resampled ensemble members quantify the complex covariance that captures both intensity and position error dynamics, without requiring additional processing on observations, meteorological fields, or other physical parameters. We tested the VTS-EnKF on two severe dust storm events that occurred in 2021. Our results show superior assimilation performance compared to the standard EnKF, particularly when position errors are present in the simulated dust plume.

**RC:**   *2) The authors have confined their consideration to the issue of dust emission uncertainty in the EnKF's P matrix, as detailed in Section 3.1. However, given that the introduction highlights the emergence of position errors as a result of meteorological input, Why not incorporate meteorological uncertainty within the process of ensemble generation?*

 AR:  Thanks for the in-depth comment. Now we have re-design our assimilation system: Each of our ensemble simulation (N=32) is driven by the perturbed emission and perturbed meteorology input. It turns out that only

trivial wilder ensemble spread found and the results are quite similar to the original EnKF experiments. We believe that the position error is mainly caused by the meteorology while ECMWF's ensemble forecast is not sufficient to account for the position error. By our method, the error can be alleviated.

Below are the new descriptions about the consideration of both emission and meteorology uncertainties:

[revised manuscript text omitted]

**RC:** *3) The decision to merge information from five distinct time points, centering around the central time, is mentioned, yet the rationale behind selecting these specific time points for combination is not fully explained. Could you please elaborate on the relevance of this choice?*

AR: Thanks for the comment. We agree that the choice of neighbouring time ensembles can affect the analysis.

[Figure]

**Figure 6.** Spatial distribution of standard deviation from model ensembles with scatter of model-minus-observation differences (absolute value) at 11:00 in DSE1(**a**) and 08:00 in DSE2(**b**). The initial assimilation analysis is performed at these time. Colorbar left is for model-minus-observation differences and right is for standard deviation.

An improper choice of interval can lead to undesirable analysis, such as less effective ensemble members (interval too small) or unrepresentative ensemble covariances (interval too large). We didn't explore the choice of neighbouring time interval in our original manuscript. Here, we design new experiments that use different time intervals to tell the impact of the choice of neighbouring time. The experiment settings and new section are added:

**Table 1.** Experiment settings.

| Name | Running ensemble number | Initial assimilation time set (hour) | Ensemble set | Localization distance (km) |
|---|---|---|---|---|
| *Control* | 32 | None | [32] | None |
| *EnKF* | 32 | $t$ | [32] | None |
| *L500* | 32 | $t$ | [32] | 500 |
| *VTS-EnKF* | 160 | $t-2, t-1, t, t+1, t+2$ | [32,32,32,32,32] | None |
| *VTS-L500* | 160 | $t-2, t-1, t, t+1, t+2$ | [32,32,32,32,32] | 500 |
| *VTS-EnKF-small* | 32 | $t-2, t-1, t, t+1, t+2$ | [6,6,8,6,6] | None |
| *VTS-L500-small* | 32 | $t-2, t-1, t, t+1, t+2$ | [6,6,8,6,6] | 500 |
| *VTS-EnKF-t1* | 96 | $t-1, t, t+1$ | [32,32,32] | None |
| *VTS-EnKF-t2* | 96 | $t-2, t, t+2$ | [32,32,32] | None |
| *VTS-EnKF-t3* | 96 | $t-3, t, t+3$ | [32,32,32] | None |
| *VTS-EnKF-t4* | 96 | $t-4, t, t+4$ | [32,32,32] | None |
| *VTS-EnKF-t5* | 96 | $t-5, t, t+5$ | [32,32,32] | None |
| *VTS-EnKF-t6* | 96 | $t-6, t, t+6$ | [32,32,32] | None |

**4.4 Sensitivity of time interval**

Previous researches have found that an improper neighboring time interval $\tau$ can lead to undesirable results, such as less-effective ensemble members (interval too small) ($\tau$ too small) or ensemble member clustering and unrepresentative ensemble covariances ($\tau$ too large) (Xu et al., 2008; Gasperoni et al., 2022, 2023). To explore the sensitivity of the choice of neighboring time interval, series of VTS-EnKF experiments with different neighboring time interval were carries out. Time intervals ranging from 1 to 6 hour were tested. As shown in Fig. 10, snapshots from 6 experiments on DSE1 clearly depicts the trend. In general, all the VTS-EnKF experiments show better performance than EnKF. While in terms of specific time interval, different patterns can be noticed. For short intervals including 1 and 2 hour, there is not sufficient ensemble spread to account for the position error. Thus there are still position error remaining and RMSE is still high. For long intervals including 5 and 6 hour, dust plume is clustered away from central dust plume. Three dust branches are noticed in *VTS-EnKF-t5* and an overly backwards dust plume is noticed in *VTS-EnKF-t6*. In this case, 3-hour interval is the best choice with the lowest RMSE (696.11 μg m$^{-3}$) and NMB (-63.5 %).

Same experiments on DSE2 are also performed and snapshots are shown in Fig. S3. Similar patterns are found on DSE2. Lowest RMSE and NMB are achieved in *VTS-EnKF-t4*. Too short interval leads to inability in position error correction and too long interval leads to excessive dust plume. Considering both cases, 3-hour interval is the preferred choice which holds the capability to handle position and not creates excessive clustered dust plume.

[Figure]

**Figure 10.** Spatial distribution of ground-based BR-PM$_{10}$ observations (scatter) and simulated dust plume (SDP) on surface from the posteriori SDP updated by VTS-EnKF-t1 (**a**), the posteriori SDP updated by VTS-EnKF-t2 (**b**), the posteriori SDP updated by VTS-EnKF-t3 (**c**), the posteriori SDP updated by VTS-EnKF-t4 (**d**), the posteriori SDP updated by VTS-EnKF-t5 (**e**), the posteriori SDP updated by VTS-EnKF-t6 (**f**) at 11:00, 15th March 2021 (CST).

[Figure]

**Figure S3.** Spatial distribution of ground-based BR-PM$_{10}$ observations (scatter) and simulated dust plume (SDP) on surface from the posteriori SDP updated by VTS-EnKF-t1 (**a**), the posteriori SDP updated by VTS-EnKF-t2 (**b**), the posteriori SDP updated by VTS-EnKF-t3 (**c**), the posteriori SDP updated by VTS-EnKF-t4 (**d**), the posteriori SDP updated by VTS-EnKF-t5 (**e**), the posteriori SDP updated by VTS-EnKF-t6 (**f**) at 11:00, 28th March 2021 (CST).

**RC:** *4) It would be helpful to clarify the methodology for detecting the occurrence of position error, especially in light of the rapid evolution of dust storms. Is it automatically detected or manually chosen? What criteria do the authors employ to determine the appropriate timing for implementing the NTEnKF?*

AR: Thanks for the comment. In this paper, we decide the timing manually. To better identify the emergence of position error, a simple identification index is designed. Descriptions are made in Supplementary. Details are shown below:
* * *
**2. Identification of position error**

To objectively identify the position error, a simple identification index is designed, which is the interquartile range of $\mathcal{H}x - y$. This index is often used to describe the spread of the data. Here, it depicts the error statistics transiting from Gaussian to non-Gaussian distribution with emergence of position error:

$$\text{IQR} = Q_3(\mathcal{H}x - y) - Q_1(\mathcal{H}x - y) \qquad (1)$$

IQR is referred to as interquartile range. $Q_3$ is the third quartile (75 % of the data) and $Q_1$ is the first quartile (25 % of the data).

Figure S1 is the time series of the IQR in two cases. It can be clearly seen in both cases that the IQR increases dramatically with the long -term transport of dust. It is a sign that the mismatch between model and observation (position error) is becoming obvious.
* * *
[Figure]

**Figure S1.** Time series of interquartile range in DSE1 (**a**) and DSE2 (**b**).

**RC:** *5) In both Figure 4 (b.2) and Figure 7 (b.2), there are conspicuously high values located to the west of the dust plumes following the NTEnKF analysis. I'm curious if, in the absence or scarcity of observations, applying the NTEnKF could lead to the generation of false or overly extensive dust plumes, potentially exacerbating the inaccuracies of the original model simulation.*

AR: Thanks for the comment. We agree that it is possible that EnKF with VTS exacerbate the dust plume in the absence of surrounding observations. Therefore, we only used neighbouring $\pm 1$ and $\pm 2$ hour to weaken the impact. Sensitivity tests concerning the neighbouring time interval choice is presented in Comment 3).

> In particular, the western extent of the dust plume is covered by the insufficient stations, which results in an inadequate representation of the dust load. By incorporating neighbouring ensembles, the dust plume is extended wilder, as can't be verified by the observations.

**RC:** *6) The scatters depicted in all the figures are too small to recognize, making it challenging for readers to quickly grasp the information being conveyed. Consider magnifying these visuals or narrowing the focus of the map to enhance the visibility of the dust shapes and allow for a more immediate and clear interpretation.*

AR: Thanks for the comment. We have enlarged most of the figures. It should be easier to follow now.

---

## Author Response (AR2)

**Authors' Response to Reviews of**

**Valid time shifting Ensemble Kalman filter (VTS-EnKF) for dust storm forecasting**

Mijie Pang, Jianbing Jin*, Arjo Segers, Huiya Jiang, Wei Han, Batjargal Buyantogtokh, Ji Xia, Li Fang, Jiandong Li, Hai Xiang Lin, and Hong Liao
*Geoscientific Model Development Discussions,*
* * *
**RC:** *Reviewers' Comment*,     AR: Authors' Response,     ☐ Manuscript Text

**1.  Overview**

Response to Referee #2: We would like to thank the referee for the careful review throughout the paper and the in-depth comments that help to improve our paper.

**2.  Major Comments**

**RC:**  *There many glaring language errors throughout this manuscript. These errors are likely egregious enough to distractingly irritate readers with strong English skills. Here is a list of those errors/misunderstandings in the first page alone.*

*L3: "model bias" -> "model error" DA corrects errors. Biases are a subset of errors.*

*L4: "... algorithm that effectively tunes models..." -> "algorithm that effectively improves numerical forecasts ..." When I think of "tunes model", I think of the model parameters (e.g., reaction rate coefficients) being tuned. In other words, your phrasing may mislead readers into thinking that you are doing parameter estimation (which is a separate kind of problem from your study).*

*L5-6: "However, when the position of the simulation does not align consistently with the observations which is referred to as position error, the EnKF algorithm struggles" -> "However, when the positions of simulated features are inconsistent with those observed (i.e., position errors), the EnKF algorithm struggles."*

*L7: "EnKF can hardly represent this uncertainty" -> "EnKF cannot adequately treat this error" It is the ensemble's job to represent the uncertainty, not the EnKF.*

*L8: "standard EnKF" -> "stochastic EnKF" As you are aware, there are many EnKF methods. I am unsure if the EnKF community has settled on whether the Burgers et al EnKF filter is called the "standard" EnKF. The names "stochastic EnKF" or "perturbed observation EnKF" are more precise and less likely to invite consternation.*

*L9: "ensembles" -> "ensemble members" In the ensemble DA literature, "ensemble" means "a group of model runs", and "an ensemble MEMBER" means "a model run inside the ensemble". Please use "ensemble members" to refer to having multiple model runs, and "ensemble" to refer to a group of model runs. This conflation of "ensemble" and "ensemble members" occurs throughout the manuscript.*

*L9-10: "In addition to the original ensembles quantifying dust loading variation, this methodology introduces extra ensembles from neighboring time for describing the potential spread of dust position." -> "In addition to the original ensemble members that quantify dust loading variations, this methodology introduces extra ensemble members from neighbouring valid times to incorporate position uncertainties."*

*L10-11: "allowing observations to be thoroughly resolved into the assimilation calculations" -> "thus enhancing the assimilation of observations"*

*L12: "that position error" -> "that position errors"*

*L16: "Dust storms are a natural meteorological disaster (Zhang et al., 2005) whose occurrence is attributed to frequent strong..." -> "Dust storms are natural meteorological disasters (Zhang et al., 2005) attributed to frequent strong..."*

*It is not the reviewers' job to copy-edit manuscripts. I strongly recommend engaging either an editorial service or a good writer work through the manuscript. If possible, ask a specialist on meteorological EnKFs to look over your manuscript.*

AR: Thanks for the comment. We have adopted an editorial service to edit the manuscripts. Below is the polished abstract and introduction:

**Abstract.** Dust storms pose significant risks to health and property, necessitating accurate forecasting for preventive measures. Despite advancements, dust models grapple with uncertainties arising from emission and transport processes. Data assimilation addresses these by integrating observations to rectify model error, enhancing forecast precision. The Ensemble Kalman Filter (EnKF) is a widely-used assimilation algorithm that effectively optimize model states, particularly in terms of intensity adjustment. However, the EnKF's efficacy is challenged by position errors between modeled and observed dust features, especially under substantial position errors. This study introduces the Valid Time Shifting-Ensemble Kalman Filter (VTS-EnKF) which combines stochastic EnKF with a valid time shifting mechanism. By recruiting additional ensemble members from neighboring valid times, this method not only accommodates variations in dust load but also explicitly accounts for positional uncertainties. Consequently, the enlarged ensemble better represents both the intensity and positional errors, thereby optimizing the utilization of observational data. The proposed VTS-EnKF was evaluated against two severe dust storm cases from spring 2021, demonstrating that position errors notably deteriorated forecast performance in terms of Root Mean Square Error (RMSE) and Normalized Mean Bias (NMB), impeding the EnKF's effective assimilation. Conversely, the VTS-EnKF improved both the analysis and forecast accuracy compared to the conventional EnKF. Additionally, to provide a more rigorous assessment of its performance, experiments were conducted using fewer ensemble members and different time intervals.

RC: *Your manuscript and results can be misinterpreted to mean "EnKFs cannot handle position errors". My interpretation of your results is "EnKFs can handle position errors (to some degree) IF the ensemble has sufficient spread in feature positions". The issue that you are addressing is with the ensemble used, not the EnKF algorithm itself. Please go over your manuscript and make sure that you have explicitly communicated this distinction. Perhaps all statements that "the EnKF cannot handle position errors" should be replaced with "the EnKF cannot handle position errors if the ensemble is under-dispersive with regards to feature positions". To further limit the potential for misinterpretation, please rename your "EnKF" experiment as the "Basic" or "Naive" experiment.*

AR: Thanks for the comment. We agree that EnKF is unable to accurately account for position errors when the ensemble it uses is under-dispersive in terms of the locations of features. All the statements that can be misinterpreted is replaced (text shown below is in LineXXX). The experiment *EnKF* is renamed as *Basic*.

The challenge lies in the quantification of position error and its subsequent inaccurate formulation of the background error covariance matrix. Consequently, EnKF calibrates both intensity and position error, while it cannot handle position errors if the ensemble is under-dispersive with regard to position. This deficiency curtails the capacity of current assimilation methodologies to correct position error.

RC: *The authors used ground station observations to validate their experiments and forecasts. However, as the authors have noted in L396 "the dust plume is covered by the insufficient stations". Have the authors considered using satellite imagery? The authors can acknowledge this possibility in their areas for future research.*

AR: Thanks for the comment. We recognize the current limitation in our study due to the sparse distribution of ground stations in north-west, as mentioned in Section 2.1. One promising direction for future research,

as pointed out by the reviewer, is to integrate satellite-derived dust optical depth (DOD) into our analysis. Satellite data, with its wide spatial coverage, can potentially bridge the gaps where ground observations are scarce. We acknowledge this as a valuable addition to our research agenda and plan to explore the utilization of DOD in upcoming studies.

> In particular, the western extent of the dust plume is covered by the insufficient stations, which results in an inadequate representation of the dust load. By incorporating neighboring ensemble, the dust plume is extended wilder. In the future research, assimilating satellite-derived dust optical depth (DOD) observations that have broader coverage may help to better constrain the enlarged ensemble.

**RC:** *L420-423: "The EnKF analysis, however, does not improve this dust forecast after the initial assimilation." I disagree with this statement. Figures 7a.1, 7a.2 and 7a.3 clearly show that the EnKF experiment's RMSEs are lower than the Control experiment. I also disagree that "the RMSE and NMB of the dust forecast from the EnKF scenario are nearly identical to the Control Run". Figure 7a.2 and 7a.3 indicate that the EnKF can have up to 100 ug/m3 less RMSE than the Control.*

**AR:** Thanks for the comment. In the forecast step, we use a cyclic forecast at the interval of 3-hour. In this sentence, we indicate that the EnKF analysis only didn't improve the dust forecast after the **initial** assimilation, which is the forecast starting from 2021-03-15 11:00 in Fig. 7(a.1, b.1). Hence, at the first assimilation time point, we concluded that there are trivial improvements from the EnKF to the forecast performance.

**3. Minor comments**

**RC:** *L37: "uncertain input data". Input data uncertainties are not a part result of numerical approximations. These uncertainties exist naturally on their own.*

**AR:** Thanks for the comment. We agree that uncertainties exist naturally on their own. This sentence is revised into a reasonable manner:

> However, model forecast skill is limited by the uncertain input data (e.g., wind field and boundary/initial conditions) and numerical approximations (like coarse grid cell and time step) (Mallet and Sportisse, 2006)

**RC:** *Caption of Figures 1, 4, 5 and 6: you use the phrasing "with scatters of the model-minus-observation differences" or something similar. Please be more explicit with saying that the colored circles indicate observations/model-minus observations. Readers will easily miss this very important detail. A better phrasing, for example, is "The model-minus-observation differences at various observation sites are indicated by the plotted filled circles".*

**AR:** Thanks for the comment. We admit that the original captions are implicit. They have been replaced by

> **Figure 1.** Evolution of the simulated dust plume from average of ensemble members (**a.1-3**). Their corresponding standard deviation from ensemble members (**b.1-3**) at 08:00, 11:00 and 14:00 15th March, 2021, respectively. Figures below are the same except the time is at 05:00 (**c.1** and **d.1**), 08:00 (**c.2** and **d.2**), 11:00 (**c.3** and **d.3**) 28th March, 2021, respectively. The filled circles represent ground BR-PM$_{10}$ observations in (**a**) and (**c**), and the model-minus-observation differences (absolute value) at various observation sites in (**b**) and (**d**). The colorbar in panel **a** and **c** represents the concentrations, and the colorbar in panel **b** and **d** represents the model-minus-observation differences (left) and standard deviation (right). BR-PM$_{10}$: baseline-removed PM$_{10}$.
>
> **Figure 4.** Spatial distribution of simulated dust plume (SDP) on surface from average of ensemble members at central time (**a.1**), the posteriori SDP updated by EnKF (**a.2**), the posteriori SDP updated by EnKF with localization (**a.3**), central and neighboring time ensemble model mean (**b.1**), the posteriori SDP updated by VTS-EnKF (**b.2**), the posteriori SDP updated by VTS-EnKF with localization (**b.3**) at 11:00, 15th March 2021 (CST). The filled circles are ground-based BR-PM$_{10}$ observations.

**Figure 5.** Spatial distribution of simulated dust plume (SDP) on surface from average of ensemble members at central time (**a.1**), the posteriori SDP updated by EnKF (**a.2**), the posteriori SDP updated by EnKF with localization (**a.3**), central and neighboring time ensemble model mean (**b.1**), the posteriori SDP updated by VTS-EnKF (**b.2**), the posteriori SDP updated by VTS-EnKF with localization (**b.3**) at 11:00, 28th March 2021 (CST). The filled circles are ground-based BR-PM$_{10}$ observations.

**Figure 6.** Spatial distribution of standard deviation from ensemble members at 11:00 in DSE1(**a**) and 08:00 in DSE2(**b**). The initial assimilation analysis is performed at these time. The filled circles are model-minus-observation differences (absolute value). Colorbar left is for model-minus-observation differences and right is for standard deviation.

**RC:** *L279: What do you mean by "root of error from observations"? Do you mean the "observation error standard deviation"?*

AR: Thanks for the comment. Observation error standard deviation is exactly what we mean.

**RC:** *L274-275: "Its mean is 0 and covariance is the root of diagonal from O". This description of covariance is incorrect. While we do take use square-root of O to generate the noise samples, the covariance of those noise samples is still O! You can check this for yourself in Python. The correct sentence is "Its mean is 0 and covariance is O."*

AR: Thanks for the comment. This is apparently a mistake. It is corrected in LineXXX:

Its mean is 0 and covariance is the diagonal of **O**.

**RC:** *It is not clear to me what observations are assimilated. Are you only assimilating the ground-based observation network? Please add a sentence in Section 3.3 that explicitly describes which observations you are assimialting.*

AR: Thanks for the comment. In Section 2.1, we described the observation we assimilate, which is the bias-corrected PM$_{10}$ concentrations from ground monitoring network. But we didn't mention that in Section 3.3. Now an explicit description is made in LineXXX:

The BC-PM$_{10}$ observations are assimilated.

**RC:** *L345: Please state the formula used to calculate NMB.*

AR: Thanks for the comment. We have added the formula for the evaluation metrics we used in Supporting Information.

**4. Evaluation metrics**

In this paper, Root mean square error (RMSE) and normalized mean bias (NMB) is used to evaluate the performance.

$$RMSE = \sqrt{\frac{\sum\limits_{i=1}^{m}(\boldsymbol{y_i} - \boldsymbol{\mathcal{H}x_i})^2}{m}}$$

$$NMB = \frac{\sum\limits_{i=1}^{m}(\boldsymbol{y_i} - \boldsymbol{\mathcal{H}x_i})}{\sum\limits_{i=1}^{m}\boldsymbol{y_i}}$$

$m$ is the number of observations.

**RC:** *Table 1: "Running ensemble member" – what do you mean by "Running"? Do you mean that you, for example, actually time-integrate 160 members? I am guessing that you mean "ensemble size used by the EnKF".*

AR: Thanks for the comment. It refers to both the ensemble size used by EnKF that produces analysis and ran by the model that produce forecast. We have revised it as "Ensemble size used by analysis and forecast"

RC: *L372: "imbalanced uncertainty". I appreciate that you are trying to use your own words to describe the situation. However, the ensemble DA literature has a term for this: "ensemble underdispersion".*

AR: Thanks for the comment. We have used this proper literature throughout the paper.

RC: *L430: "a reduction of RMSE is observed". A reduction relative to what? Please state that explicitly.*

AR: Thanks for the comment. We have made a more clear statement:

> By applying the VTS-EnKF analysis, a reduction of RMSE compared to the model run and EnKF can be observed in panel a. There is an approximate decrease of 100 µg m$^{-3}$ in *VTS-EnKF* compared to *Basic*, which indicates that the VTS-EnKF analysis effectively corrects the position error.

RC: *L426-427: I also disagree that "the localization method is unable to enhance the forecast". Figure 7 clearly shows that localization IS improving the forecast performance, albeit only slightly. The word "unable" is not the same as "slightly able to".*

AR: Thanks for the comment. There is only slight improvements with the localization for sure. We have used a appropriate expression on it in LineXXX:

> Moreover, it has been observed that the localization method only improves the forecast slightly in the presence of position errors.

RC: *Your use of IQR (supplement) is problematic. A Gaussian distribution with massive spread will exhibit massive IQR. In other words, IQR does not distinguish between Gaussian and non-Gaussian distributions. A simple metric that might work is: |OmB|/(obs error variance + bg error variance). |OmB| is the absolute difference between the observation and background. For Gaussian distributions, that metric rarely exceeds 3. If you consistently see values greater than 3, then you have a clear sign of non-Gaussianity.*

AR: Thanks for the comment. We admit that the IQR cannot fully describe the non-Gaussian error distribution. As the reviewer suggests, the Squared Normalized Innovation is used to better identify the non-Gaussian statistics. Detailed descriptions are made in Supplementary:

> **3. Identification of position error**
>
> To objectively identify the position error, a simple identification index is applied, which is the Squared Normalized Innovation (SNI). SNI is a measure of how well the model forecasts match the observations and is crucial for evaluating the performance of the filter and diagnosing issues related to the error distribution assumptions, particularly the assumption of Gaussian errors (Zupanski and Zupanski, 2006). Here, it depicts the error statistics transiting from Gaussian to non-Gaussian distribution with emergence of position error:
>
> $$\text{SNI} = \frac{\sum\limits_{i=1}^{m} (\boldsymbol{y}_i - \boldsymbol{\mathcal{H}} \boldsymbol{x}_i)^2}{trace(\mathbf{O}) + trace(\boldsymbol{\mathcal{H}} \mathbf{P}^f \boldsymbol{\mathcal{H}}^{\mathrm{T}})}$$
>
> $m$ is the number of observations.
>
> If the values consistently diverging from one would indicate non-Gaussian error distributions. Figure S1 is the time series of the SNI in two cases. It can be clearly seen in both cases that the SNI increases dramatically with the long -term transport of dust. It is a sign that the mismatch between model and observation (position error) is becoming obvious.

[Figure]

**Figure S1.** Time series of SNI in DSE1 (**a**) and DSE2 (**b**).

**References**

Mallet, V. and Sportisse, B.: Uncertainty in a Chemistry-Transport Model Due to Physical Parameterizations and Numerical Approximations: An Ensemble Approach Applied to Ozone Modeling, J. Geophys. Res., 111, , 2006.

Zupanski, D. and Zupanski, M.: Model Error Estimation Employing an Ensemble Data Assimilation Approach, Mon. Weather Rev., 134, 1337–1354, , 2006.

**Authors' Response to Reviews of**

**Valid time shifting Ensemble Kalman filter (VTS-EnKF) for dust storm forecasting**

Mijie Pang, Jianbing Jin*, Arjo Segers, Huiya Jiang, Wei Han, Batjargal Buyantogtokh, Ji Xia, Li Fang, Jiandong Li, Hai Xiang Lin, and Hong Liao
*Geoscientific Model Development Discussions,*
* * *
**RC:** *Reviewers' Comment*,  AR: Authors' Response,  ☐ Manuscript Text

**1. Overview**

Response to Topic editor: We would like to thank the editor for the careful review throughout the paper and the in-depth comments that help to improve our paper.

**2. Major comments**

**RC:**  *Based on section 3.1, the error covariance matrices and Kalman gain are directly calculated with the ensemble perturbations. This is different from the common used EnKF algorithms. I expect that the dimension of the dust concentration (model variable) is large given the large domain of the dust model (LOTOS-EUROS). Please clarify how the error covariance matrices are calculated and updated in this study.*

**AR:** Thanks for the comment. In fact, we didn't directly calculate the prior error covariance matrix since its large size. A more efficient way is adopted as described in Supplementary:
* * *
**2. Efficient calculation of EnKF and localization**

For the sake of computational efficiency, the explicit computation of $\mathbf{P}^f$ is eschewed given its substantial dimensionality, which entails a vast number of state covariance ($\mathbb{R}^{n \times n}$, equating to $39200 \times 39200$ in the context of our study). Instead, we opt for deriving a new perturbation matrix $\mathbf{U}$ through the following operation:

$$\mathbf{U} = \mathcal{H}\mathbf{X}^{f\prime}$$

Notably, $\mathbf{U}$ exhibits significantly reduced dimensions ($\mathbb{R}^{n \times m}$, approximately $39200 \times 1000$), thereby alleviating the burden compared to the background error covariance matrix $\mathbf{P}^f$.

Subsequently, by incorporating $\mathbf{U}$ into the calculation of the Kalman gain $\mathbf{K}$, we obtain:

$$\mathbf{K} = \frac{1}{N-1}\mathbf{X}^{f\prime}\mathbf{U}^{\mathrm{T}}\left(\frac{1}{N-1}\mathbf{U}\mathbf{U}^{\mathrm{T}} + \mathbf{R}\right)^{-1}$$

This formulation enables a scalable reduction in both memory requirements and computational expenses, tailored to the dimensions of the model state and the observational dataset.

To further enhance computational practicality, the localization scheme is applied to $\mathbf{K}$ as follows:

$$\mathbf{K} = \frac{1}{N-1}\boldsymbol{\rho}_{n \times m} \circ \mathbf{X}^{f\prime}\mathbf{U}^{\mathrm{T}}\left(\frac{1}{N-1}\boldsymbol{\rho}_{m \times m} \circ \mathbf{U}\mathbf{U}^{\mathrm{T}} + \mathbf{R}\right)^{-1}$$

Here, $\boldsymbol{\rho}_{n \times m}$ signifies the cross-correlation between model states and observations, while $\boldsymbol{\rho}_{m \times m}$ denotes the autocorrelation among observations (Evensen et al., 2022), thereby embodying an advanced strategy for managing the spatio-temporal correlations inherent in complex systems.
* * *
**RC:** *After the VTS-EnKF update, will the enlarged ensemble size be reduced to the original size? How were the ensemble members selected?*

**AR:** Thanks for the comment. As Fig. 3 shows, VTS-EnKF is applied only in the initial assimilation analysis. Then the forecasts are made by the enlarged ensemble. The enlarged ensemble size won't be reduced to the original size. The cyclic forecasts are made on these ensemble members. In practice, smaller number of central time ensemble members can be set to be less computation-demanding and achieve adequate performance with aid of localization as described in Sect. 4.3.

[Figure]

**Figure 3.** Sequential assimilation time set for DSE1 (**a**) and DSE2 (**b**). Take DSE1 for instance, the assimilation analysis is performed at the intervals of 3 hours from 11:00 to 17:00 and the rolling forecast is made with a horizon of 24 hours based on the assimilation analysis. The EnKF with VTS and EnKF is performed in turn.

**RC:** *The emission parameterization should be briefly mention in section 2 and clarify what factors in the parameterization contributes to the uncertainty.*

**AR:** Thanks for the comment. More detailed description about the emission parameterization is added Supplementary:
* * *
**5. Emission parameterization**

The dust flux rate $f$ is calculated as a function of horizontal saltation $F_h$, the sandblasting efficiency $\alpha$, a terrain preference $S$, and an erodible surface fraction $C$ as:

$$f = F_h \times \alpha \times S \times C$$

The horizontal saltation $F_h$ represents the horizontal flux rate, which is proportional to the third power of the wind friction velocity $u_*$, as long as this exceeds a certain friction velocity threshold $u_{*t}$. Explicitly, $F_h$ in a given grid cell is computed from:

$$F_h = \begin{cases} 0, & u_* \leq u_{*t} \\ \frac{\rho_a}{g} u_*^3 (1 + \frac{u_{*t}}{u_*})(1 - \frac{u_{*t}^2}{u_*^2}), & u_* > u_{*t} \end{cases}$$
* * *
where $g$ denotes the gravitational constant, and $\rho_a$ represents the atmospheric density. The friction velocity $u_*$ is computed from the ECMWF wind speed at 10 m height assuming neutral atmospheric stability, following a logarithmic profile. The friction velocity threshold (FVT) $u_{*t}$ represents the minimum friction velocity to initiate the movement of soil particles.

Emission errors are also likely to be induced during the formation of the friction velocity ($u_*$) from meteorology data, the terrain preference ($S$) from the topography resource, and the erodible surface fraction ($C$) from the land cover database. Among these factors, the friction velocity threshold (FVT, $u_{*t}$) is very important and sensitive for the outcome, since it directly influences whether dust saltation will occur and also quantifies the amplitude of the flux rate.

**3. Minor comments**

**RC:** *Please use the formal format for the time (e.g. 1100 UTC).*

AR: Thanks for the comment. China Standard Time (CST) is the local time of where our study mainly focused on. We used it for the consistency with our previous research and better demonstrates the impact to the local region. We have added a description of this time format to the captions to avoid confusion.

**RC:** *In the abstract: Even with position error, the perturbations can be Gaussian. I would remove the sentence "EnKF can be bias for the non-Gaussian statistics", since this sentence is irrelevant to this study.*

AR: Thanks for the comment. We admit that this sentence is excessive in the abstract. It is removed in the revised version.

**RC:** *L71, Page 3: "For non-Gaussian problem...". Please revise this sentence.*

AR: Thanks for the comment. This sentence is revised as:

> Despite these strengths, the EnKF, as an extension of the Kalman Filter, presumes Gaussian error distributions (Amezcua and Van Leeuwen, 2014). When dealing with non-Gaussian error statistics, EnKF can create suboptimal outcomes for the linearized dynamics or operators and sampling errors caused by finite ensemble members (Lei et al., 2010).

**RC:** *L385: This sentence is confusing. Please make it clearer.*

AR: Thanks for the comment. In this sentence, we intend to describe that the uncertainty spread has expanded by applying VTS. The ensemble underdispersion is alleviated, thus enhance the capability of EnKF to deal with the position errors. It is revised in LineXXX:

> This expansion of the uncertainty spread effectively addresses the issue of ensemble underdispersion, thereby boosting the EnKF's capability to handle position errors.

**References**

Amezcua, J. and Van Leeuwen, P. J.: Gaussian Anamorphosis in the Analysis Step of the EnKF: A Joint State-Variable/Observation Approach, TELLUS A, 66, 23 493, , 2014.

Evensen, G., Vossepoel, F. C., and van Leeuwen, P. J.: Data Assimilation Fundamentals: A Unified Formulation of the State and Parameter Estimation Problem, Springer Textbooks in Earth Sciences, Geography and Environment, Springer International Publishing, Cham, , 2022.

Lei, J., Bickel, P., and Snyder, C.: Comparison of Ensemble Kalman Filters under Non-Gaussianity, Mon. Weather Rev., 138, 1293–1306, , 2010.

---

## Author Response (AR3)

**Authors' Response to Reviews of**

**Valid time shifting Ensemble Kalman filter (VTS-EnKF) for dust storm forecasting**

Mijie Pang, Jianbing Jin*, Arjo Segers, Huiya Jiang, Wei Han, Batjargal Buyantogtokh, Ji Xia, Li Fang, Jiandong Li, Hai Xiang Lin, and Hong Liao
*Geoscientific Model Development Discussions,*
* * *
RC: *Reviewers' Comment*,     AR: Authors' Response,     □ Manuscript Text

**1. Overview**

Response to Editor: We would like to thank the editor for the careful review throughout the whole revision process and all the in-depth comments that help to improve our paper.

**2. Comments**

RC:    *N is defined later (Line 197). The sentence starting with "N=32" is informal. Please revise this sentence.*

AR:    Thanks for the comment. The definition here is removed. The formal definition is put in the following sentence.

> $$[\boldsymbol{x}_1, \, ... \, , \boldsymbol{x}_\mathrm{N}] \; = \; [\mathcal{M}(\boldsymbol{f}_1, \boldsymbol{w}_1), \, ... \, , \, \mathcal{M}(\boldsymbol{f}_\mathrm{N}, \boldsymbol{w}_\mathrm{N})]$$
>
> N refers to the total ensemble number, and the choice will be explained in Section 3.3.

RC:    *Line 199: This sentence should be revised to something like "the ECMWF ensemble forecasts are re-gridded to match the model resolution".*

AR:    Thanks for the comment. This sentence is revised as:

> The 6-hourly short-term meteorological forecast field is interpolated to hourly values and re-gridded to match the model resolution.

RC:    *Line 197 and 202 are the same? I suggest to combine these two sentences.*

AR:    Thanks for the comment. The former sentence is deleted to avoid repetition.

> Meteorologic field $[\boldsymbol{w}_1, ..., \boldsymbol{w}_\mathrm{N}]$ are randomly selected from the total 51 ensemble meteorology.

RC:    *Line 230: I suggest removing "pure" since "the pure model forecast" is confusing.*

AR:    Thanks for the comment. "pure" is removed throughout the manuscript.

> This position error not only limits the model forecast performance but also significantly degrades the subsequent assimilation analysis and forecast.

RC:    *Line 287: The covariance of the sampling error should be the same as O.*

AR:    Thanks for the comment. We meant to refer the variance is the root of diagonal of **O**. Thanks for point out this mistake.

$\epsilon^i$ represents the sampling error vector. It is a random vector subjecting to normal distribution. Its mean is 0 and variance is the root of diagonal from **O**.

**RC:** *Line 467: Please briefly clarify how are the eight central ensemble members selected?*

AR: Thanks for the comment. The sentence is revised as:

These experiments start from 8 ensemble members that are driven by randomly selected emission and meteorology field from the origin ensemble. During the initial assimilation, the extra 4×6 ensemble members from neighboring ±1 and ±2 hours are randomly sampled from these 8 ensemble members. The new ensemble comprises 32 members which is equivalent to the origin ensemble number of *Basic*.

**References**

**Authors' Response to Reviews of**

**Valid time shifting Ensemble Kalman filter (VTS-EnKF) for dust storm forecasting**

Mijie Pang, Jianbing Jin*, Arjo Segers, Huiya Jiang, Wei Han, Batjargal Buyantogtokh, Ji Xia, Li Fang, Jiandong Li, Hai Xiang Lin, and Hong Liao
*Geoscientific Model Development Discussions,*
* * *
RC: *Reviewers' Comment*,    AR: Authors' Response,    ☐ Manuscript Text

**1. Overview**

RC:   *The authors have done an excellent job in revising their manuscript. Great job! The manuscript reads well and I am happy with their extensive sensitivity tests. As such, I recommend accepting this manuscript for publication with one minor edit.*

Response to Reviewer #1: We would like to thank the reviewer for the careful review throughout the whole revision process and all the in-depth comments that help to improve our paper.

**2. Minor comment**

RC:   *In the paragraph starting at line 289, you highlighted the systematic position errors in the free run. That seems out of place in a section that focuses on describing your methodology. Perhaps you should shift that discussion to section 4.1.*

AR:   Thanks for the comment. We have re-organized these paragraphs to Section 4.1 to make the discussion more coherent.

> **4.1 Impact on assimilation analysis**
>
> There are noticeable position errors arise with the transport of dust storm. It is clearly shown in Fig. 1 (b,d) that the spatial distribution of the standard deviation (square root of the diagonal values in $\mathbf{P}^f$) from 32 model ensemble members, along with the scatter of absolute model-minus-observation differences in two cases (DSE1, DSE2). In general, their spatial distribution corresponds well to the simulated dust field depicted in Fig. 1 (a, c). Concurrently, the uncertainty in the light blue box decreases rapidly as the simulated dust plume moves southward, as illustrated in panels b.1 and b.2. This suggests that our ensemble model simulations are highly confident that there are less affected by dust aerosols. However, the observations indicate that this area remains heavily polluted. In the case of DSE2, the situation becomes more complex. The simulated dust plume in DSE2 covers most of the observation area with a high dust load, as demonstrated in panels c.1 and d.1. The uncertainty, on the other hand, reveals that the ensemble model is less confident about the dust load, especially in the light blue box displayed in panel d.2. After 3 hours, these discrepancies become more evident. The extent to which this situation affects the EnKF assimilation will be discussed in this paper. It poses a challenge to EnKF assimilation in resolving the high-value measurements in this region.
>
> Subsequent results have confirmed this theory. Figure 4 displays the spatial distribution of ground BR-PM$_{10}$ observations (scatter) and dust field forecasts from the average of the ensemble (panel a.1), the posteriori from EnKF analysis (panel a.2) and EnKF with localization (panel a.3), the average of the enlarged ensemble (panel b.1), the posteriori from VTS-EnKF analysis (panel b.2) and VTS-EnKF analysis with localization (panel b.3) at 11:00, 15th March, 2021 China Standard Time (CST). It should be noted that the average dust concentrations in panel b.1 are calculated from the 160 ensemble simulations used in VTS-EnKF, which slightly differ from the average of 32 ensemble members. In

DSE1, the RMSE and NMB from the pure ensemble model simulation are as high as 856.36 µg m$^{-3}$ and -78.31 %. Both EnKF and LEnKF assimilation analyses achieve very limited improvement in estimating the dust state field. As shown in panel a.2 and panel a.3, the RMSE and NMB remain high at 819.04 µg m$^{-3}$ and -75.65 % in *Basic*, and 782.57 µg m$^{-3}$ and -73.52 % in *L500*. The main reason for this is the ensemble underdispersion, as described in Sect. 3.2. As observed in the light blue box in panel a.1, the simulated dust plume is located farther southeast compared to the PM$_{10}$ measurements. This snapshot exhibits an apparent position error. After EnKF analysis, the simulated dust plume in the light blue box barely changes, as depicted in panel a.2. Numerous ground stations in this area report high PM$_{10}$ concentrations, but the assimilated dust field fails to resolve most of them. The localization method offers limited assistance in this situation, as illustrated in panel a.3. With the unresolved positional error, the EnKF, which focuses more on intensity correction, is much less effective.

**References**